# Proteomic characteristics reveal the signatures and the risks of T1 colorectal cancer metastasis to lymph nodes

Aojia Zhuang[1†], Aobo Zhuang[1,2†], Yijiao Chen[1,3†], Zhaoyu Qin[1†], Dexiang Zhu[1,3], Li Ren[1,3], Ye Wei[1,3], Pengyang Zhou[1,3], Xuetong Yue[1], Fuchu He[1,4,5*], Jianmin Xu[1,3*], Chen Ding[1,6*]

[1]State Key Laboratory of Genetic Engineering and Collaborative Innovation Center for Genetics and Development, Human Phenome Institute, School of Life Sciences, Institutes of Biomedical Sciences, Department of Colorectal Surgery, Colorectal Cancer Center, Zhongshan Hospital, Fudan University, Shanghai, China; [2]Xiamen University Research Center of Retroperitoneal Tumor Committee of Oncology Society of Chinese Medical Association, College of Medicine, Xiamen University, Xiamen, China; [3]Cancer Center, Zhongshan Hospital, Fudan University, Shanghai, China; [4]State Key Laboratory of Proteomics, Beijing Proteome Research Center, National Center for Protein Sciences, Beijing, China; [5]Research Unit of Proteomics Driven Cancer Precision Medicine, Chinese Academy of Medical Sciences, Beijing, China; [6]State Key Laboratory of Cell Differentiation and Regulation, Henan International Joint Laboratory of Pulmonary Fibrosis, Henan Center for Outstanding Overseas Scientists of Pulmonary Fibrosis, College of Life Science, Institute of Biomedical Science, Henan Normal University, Xinxiang, China

*For correspondence:
hefc@nic.bmi.ac.cn (FH);
xujmin@aliyun.com (JX);
chend@fudan.edu.cn (CD)

†These authors contributed equally to this work

**Abstract** The presence of lymph node metastasis (LNM) affects treatment strategy decisions in T1NxM0 colorectal cancer (CRC), but the currently used clinicopathological-based risk stratification cannot predict LNM accurately. In this study, we detected proteins in formalin-fixed paraffin-embedded (FFPE) tumor samples from 143 LNM-negative and 78 LNM-positive patients with T1 CRC and revealed changes in molecular and biological pathways by label-free liquid chromatography tandem mass spectrometry (LC-MS/MS) and established classifiers for predicting LNM in T1 CRC. An effective 55-proteins prediction model was built by machine learning and validated in a training cohort (N=132) and two validation cohorts (VC1, N=42; VC2, N=47), achieved an impressive AUC of 1.00 in the training cohort, 0.96 in VC1 and 0.93 in VC2, respectively. We further built a simplified classifier with nine proteins, and achieved an AUC of 0.824. The simplified classifier was performed excellently in two external validation cohorts. The expression patterns of 13 proteins were confirmed by immunohistochemistry, and the IHC score of five proteins was used to build an IHC predict model with an AUC of 0.825. RHOT2 silence significantly enhanced migration and invasion of colon cancer cells. Our study explored the mechanism of metastasis in T1 CRC and can be used to facilitate the individualized prediction of LNM in patients with T1 CRC, which may provide a guidance for clinical practice in T1 CRC.

## Editor's evaluation

This paper seeks to answer an important clinical question by coming up with novel predictive biomarkers to predict high-risk T1 colorectal cancers that are at risk for nodal involvement with a

machine-learning approach. The findings underscore that T1 CRC may have unique features and pathways that contributed to LNM.

## Introduction

CRC is the third most common cancer worldwide and the third leading cause of cancer-related deaths in Western countries (*Malvezzi et al., 2014*; *Hori et al., 2015*; *Jemal et al., 2017*). With the introduction of population-based screening programs, a growing number of early invasive CRCs (T1 CRCs) are being diagnosed (*Zauber et al., 2012*). It is estimated that the endoscopic removal of adenomatous polyps can reduce CRC-related mortality by more than 50%. Evidence suggests that ESD (endoscopic submucosal dissection) alone is an effective option for T1 CRC patients, who are at low risk for developing LNM, while more extensive radical surgery after ESD is needed for only high-risk patients (*Ikematsu et al., 2013*; *Yoda et al., 2013*). According to the current clinical treatment guidelines, which rely on a histopathological examination, approximately 70 to 80% of patients are classified as high risk (poor tumor differentiation, lymphatic/vascular invasion, and depth of submucosal invasion >1000 mm). However, the LN status of only 8 to 16% of patients can be accurately predicted by these guidelines; thus, a large number of LNM-negative patients routinely undergo unnecessary additional surgeries, with an associated postoperative mortality rate of 3–6% (*Tanaka et al., 1995*; *Kobayashi et al., 2011*). The contradiction between the high rate of additional surgical resection and the fact that only a few people have LNM is due to the lack of accurate diagnostic methods. Therefore, there is an urgent need to develop a new method that can effectively determine LNM in T1 CRC.

Recent studies have focused on accurately predicting lymph node metastasis in T1CRCs to provide references for further postoperative treatment of ESD patients. Unfortunately, current research still has limitations. Previously, Ozawa et al., and Kandimalla et al., used the microRNA (miRNA) and messenger RNA (mRNA) expression dataset of T1 and T2 CRC patients from The Cancer Genome Atlas (TCGA) as the training cohort to build predictive models, achieved an area under the ROC curve (AUC) values of 0.74 and 0.84, respectively (*Ozawa et al., 2018*; *Kandimalla et al., 2019*). After that, Wada et al., validated these miRNA and mRNA signatures in the blood samples (*Wada et al., 2021*). These studies unraveled a new paradigm for a more adequate risk assessment and identification of patients who are true candidates for endoscopic treatment or radical surgery. However, these retrospective studies have several limitations, less than 10% of T1 colorectal cancer samples had lymph node metastasis, and almost no specimens removed by ESD were included. Most recently, Kudo et al., developed an algorithm to predict LNM in 4073 patients with T1 CRC by clinicopathological characteristics and achieved an AUC of 0.83 (*Kudo et al., 2021*). The accurate prediction of LNM in T1 CRC is a crucial but difficult challenge for scientists worldwide.

Proteins, as executors of biological functions, are receiving much research attention. Several recent proteomic studies of CRC defined new protein signatures, molecular subtypes, and metastasis markers (*Vasaikar et al., 2019*; *Zhang et al., 2014*; *Mikula et al., 2011*; *Saleem et al., 2019*; *Anwaier et al., 2022*; *Du et al., 2022*; *Gao et al., 2022*) and revealed differences in tumorigenesis between right- and left-sided CRC (*Yaeger et al., 2018*). However, these studies focused on advanced-stage CRC rather than early-stage CRC, and proteomic studies of T1NxM0 CRC are still lacking. There was a recent proteomic study using 21 T1 and T2 CRC patients (*Steffen et al., 2021*), however, the sample size of the study is limited.

Here, we assembled three cohorts of patients with or without LNM (a training cohort, an ESD validation cohort, and a prospective validation cohort). A quantitative proteomics approach was used to analyze a total of 221 patients. We developed a high-performance prediction model that can help to reduce the number of unnecessary additional surgeries and will benefit the majority of patients.

## Results

### Cohort characteristics and research design

To identify LNM mechanisms and protein signatures for T1NxM0 CRC, we performed mass spectrometry (MS)-based proteomics to analyze FFPE tumor samples from 143 LNM-negative and 78 LNM-positive patients with T1 CRC, totaling 221 individuals. The patients consisted of a training cohort and two different validation cohorts (*Figure 1A*). The clinicopathologic characteristics of the patients

**eLife digest** Most patients with early-stage colorectal cancer can be treated with a minimally invasive procedure. Surgeons use a flexible tool to remove precancerous or cancerous cells, cutting the risk of death from colorectal cancer in half. But a small number of early-stage colorectal cancer patients are at risk of their cancer spreading to the lymph nodes. These patients need more extensive surgery. Clinicians use risk stratification tools to decide which patients need more extensive surgery.

Unfortunately, the existing risk stratification tools are not very accurate. The current approach, which analyzes colon tissue for cancerous changes, classifies 70% to 80% of early-stage colorectal cancer patients as high risk for cancer spread. But only about 8% to 16% of patients in the high risk group have lymph node metastasis. As a result, many patients undergo unnecessary, invasive surgery.

Zhuang, Zhuang, Chen, Qin, et al. developed a more accurate way to predict which patients are at risk of lymph node metastasis using proteins. In the experiments, the team analyzed the proteins in tumor samples from 143 patients with early colorectal cancer who did not have lymph node metastases and 78 patients with metastases. Zhuang et al. then used machine learning to build a prediction tool that used 55 proteins to identify patients at risk of metastases. The new approach was more accurate than existing tools and simplified versions with only nine or five proteins also performed better than existing tools.

This work provides preliminary evidence that protein-based models using as few as five proteins can more accurately identify which patients are at risk of metastasis. These models may reduce the number of patients who undergo unnecessary invasive surgery. The experiments also identified potential targets for therapies to prevent or treat lymph metastases. For example, they showed that low levels of the RHOT2 protein predict metastasis.

are listed in *Figure 1B* and *Figure 1—source data 1*. As previously reported in T1NxM0 CRC, LNM is significantly associated with the submucosal invasion depth (p=0.014, chi-square test), lymphatic or vascular invasion (p<0.001, chi-square test), and poor differentiation (p=0.004, chi-square test) (*Figure 1—source data 1*). In our cohort, the rate of LNM was related to tumor location, specifically. The left-sided tumors (LNM rate: 41.1%) showed a higher metastatic tendency than right-sided (LNM rate: 22.0%) (p=0.036, chi-square test) (*Figure 1—figure supplement 1*).

Of the 172 patients for whom MSI status was available, 12 (7%) were MSI-H, which is consistent with previous reported on metastatic CRC by Federico et al., (6.6%) (*Innocenti et al., 2019*). (*Figure 1B* and *Figure 1—source data 1*). For MSI-H patients, there were 16.7% (2 of 12) had LNM, and 39.6% (63 of 159) for MSS patients, the MSS group showed a higher tendency of LNM (*Figure 1—figure supplement 1K*). Our results are in agreement with the previous studies (*Kang et al., 2015*).

The mutations of *RAS*, *BRAF*, and *PIK3CA* genes were also detected using PCR. The results showed that 70 of 132 (53%) individuals had no mutations in all three genes (*Figure 1B* and *Figure 1—source data 1*). Nineteen of 62 (30.6%) patients with gene mutations and 26 of 70 (37.1%) patients without mutations had LNM, indicating there was no statistical difference between the two groups (p=0.432). The mutation ratio was similar to previous studies (*Rui et al., 2015*). There were 48 (36.4%) individuals had mutations in the *KRAS*-exon2; four (3%) had *KRAS*-exon3 mutations; two (1.5%) had *KRAS*-exon4 mutations; two (1.5%) had *NRAS* mutations; four (3%) had mutations in *BRAF V600E*; three (2.3%) had *PIK3CA* mutations. There was one patient has mutations in both *NRAS*, *PIK3CA*, and *KRAS*-exon2.

For the proteomic analysis, proteins were obtained from FFPE tumor samples after cleavage with trypsin and analyzed by high-resolution LC-MS/MS on a Q Exactive HF-X mass spectrometer using a label-free technique. Overall, we produced a high-quality dataset (*Figure 1—figure supplement 2*), more than 13,000 protein groups (with a 1% false discovery rate [FDR] at the peptide and protein levels) were identified, and the identification number of each sample was over 4000 proteins (*Figure 1—source data 2*). For the bioinformatic analysis, we further filtered the data, as shown in *Figure 1—figure supplement 2H* (*Figure 1—source data 3*). In sum, we have made a comprehensive proteomic study of T1 CRC and provide a reliable data source for future research.

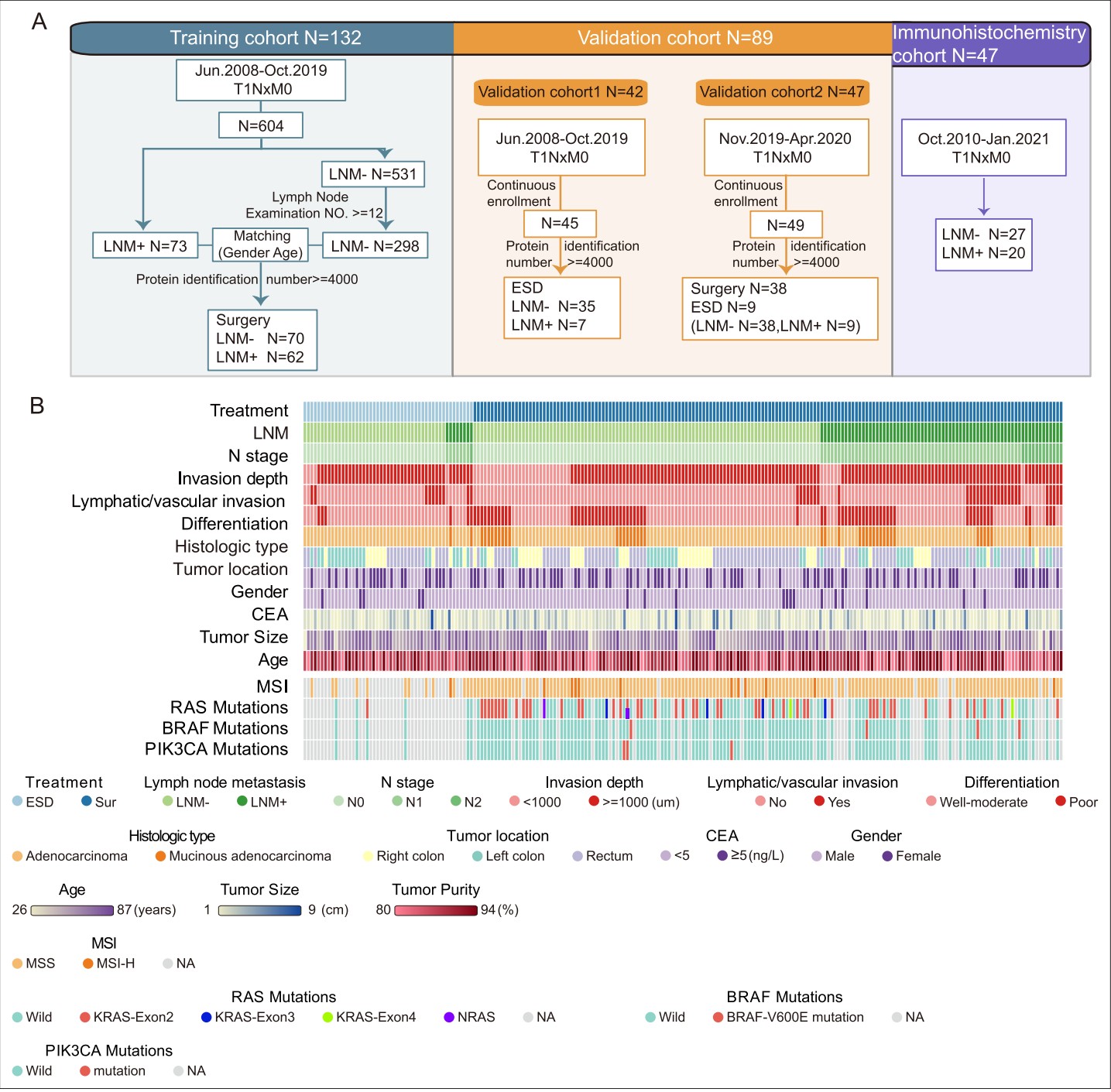

**Figure 1.** Sample selection and proteomics landscape of T1 colorectal cancer (CRC) with or without lymph node metastasis (LNM). (**A**) In total, 221 samples were divided into three cohorts: a training cohort (N=132), validation cohort 1 (N=42), and validation cohort 2 (N=47); 47 samples were used for immunohistochemistry (IHC) staining. (**B**) The study included 143 LNM-negative and 78 LNM-positive patients with T1 CRC and 51 and 170 patients treated with endoscopic submucosal dissection (ESD) or surgical resection, respectively. Clinical parameters are shown in the heatmap. Also, see *Figure 1—figure supplement 1*.

The online version of this article includes the following source data and figure supplement(s) for figure 1:

**Source data 1.** Clinicopathologic features.

**Source data 2.** All identified proteins.

**Source data 3.** Filtered proteomics data.

*Figure 1 continued on next page*

**Figure supplement 1.** Association between clinical parameters and lymph node metastasis (LNM).

**Figure supplement 2.** Quality assessment of proteomic data and summary of the analysis.

## Proteomic characteristics of the LNM-negative and LNM-positive groups

To identify protein signatures and pathways associated with LNM in T1NxM0 CRC, we investigated the differential proteomic patterns between T1 CRC patients with or without LNM. First, we surveyed the published CRC LNM markers revealed in the literature. Among the 44 reported gene or RNA markers, only CTSD, GSTM3, and MX1 (*Oh-e et al., 2001*; *Meding et al., 2012*; *Croner et al., 2014*) were differentially expressed between LNM-negative and LNM-positive patients in our proteomic data (p<0.05, Wilcoxon rank-sum tests) (*Figure 2A*). It indicates that the existing research, including animal/cell models, gene/RNA-related studies, etc., may not reflect the status of lymph node metastasis in T1 CRC. We also compared our data with those of the previous all stage CRC proteomic studies (CPTAC cohort and mCRC cohort) (*Zhang et al., 2014*; *Li et al., 2020b*), 4634 proteins were found in all three cohorts, whereas 2577 proteins were detected specifically in our T1 CRC cohort (*Figure 2—figure supplement 1A*). These results suggest that compared with advanced CRC, T1 CRC might have its own unique protein patterns. These markers might improve the understanding of LNM in early-stage CRC.

Thus, we comprehensively compared protein signatures and biological differences between the LNM-negative and LNM-positive patients with T1 CRC. We found that 82 and 84 proteins were significantly differentially expressed in LNM-negative and LNM-positive patients, respectively (identified in at least 30% of samples with a log2-fold change [log2FC]>1 or <-1 and p<0.05, Wilcoxon rank-sum test) (*Figure 2B* and *Figure 2—source data 1*). To search for druggable targets in LNM-positive T1 CRC, we investigated 84 proteins that were overrepresented in LNM-positive patients and identified 19 US Food and Drug Administration (FDA)-approved drug targets: F13A1, GBA, GNG2, GUCY1B3, STK3, STK4, VWF, etc. (*Figure 2C* and *Figure 2—source data 2*). Furthermore, 34 of these proteins, including ATAD2, BAIAP2, CEACAM6, FARS2, MX2, OSBPL5, SERPINB5, SERPINB8, SHC1, UBE2Z, YAP1, and ZG16, etc. were identified as potential drug targets (*Figure 2C*; *Freshour et al., 2021*). These potential druggable markers might provide insights into new precision medicine for T1NxM0 CRC and further benefit targeted treatment.

To further explore the biological processes associated with LNM in T1 CRC, we conducted gene set enrichment analysis (GSEA) to identify enriched pathways. The results revealed that only the epithelial-mesenchymal transition (EMT) pathway was significantly enriched (adjusted p-value <0.05) in the LNM-positive group, while no significantly enriched pathways in the LNM-negative group (*Figure 2D*, *Figure 2—source data 3*). We also examined the expression of EMT-related proteins and found that all of them were up-regulated in the LNM-positive group compared to the LNM-negative group (log2FC >0; p<0.05, Wilcoxon rank-sum tests), with six proteins showing a significant upregulation (log2FC >1), including CAP2, PLOD1, SERPINE2, etc. (*Figure 2E*).

To verify whether these EMT-associated proteins are unique to T1 CRC, we then employed two published in all stages (T1-T4) CRC datasets, and compared the EMT marker expression between the LNM-positive and LNM-negative groups. We found that the EMT markers that were significantly up-regulated (log2FC >1) in the T1 CRC cohort showed no differences in either the CPTAC or mCRC cohorts (*Figure 2F–I* and *Figure 2—figure supplement 1B and C*). However, when we focused on the early-stage (T1/T2) CRC individuals from the mCRC cohort and compared the EMT marker expression between the LNM-positive and LNM-negative groups, the EMT markers like SERPINE2 and LRP1 were significantly upregulated in both our cohort and the early-stage CRC group from the mCRC cohort (log2FC >1 and p<0.05, Wilcoxon rank-sum test) (*Figure 2F and G*). Additionally, PLOD2 was also found to be up-regulated in the LNM-positive group in T1/2 patients in the mCRC cohort. (*Figure 2H*). In summary, EMT might play an essential role during the process of LNM of tumor cells in early-CRC.

As expected, cytoskeletal remodeling proteins are differently expressed between the two groups. Cytoskeletal organization plays an important role in EMT. Cytoskeletal remodeling proteins, including ABI1, SPTA1, SPTB, ANK1, MRPL46, and ITGA2B were down-regulated in patients with LNM compared with those without LNM (log2FC <-1 and p<0.05, Wilcoxon rank-sum test). Previous

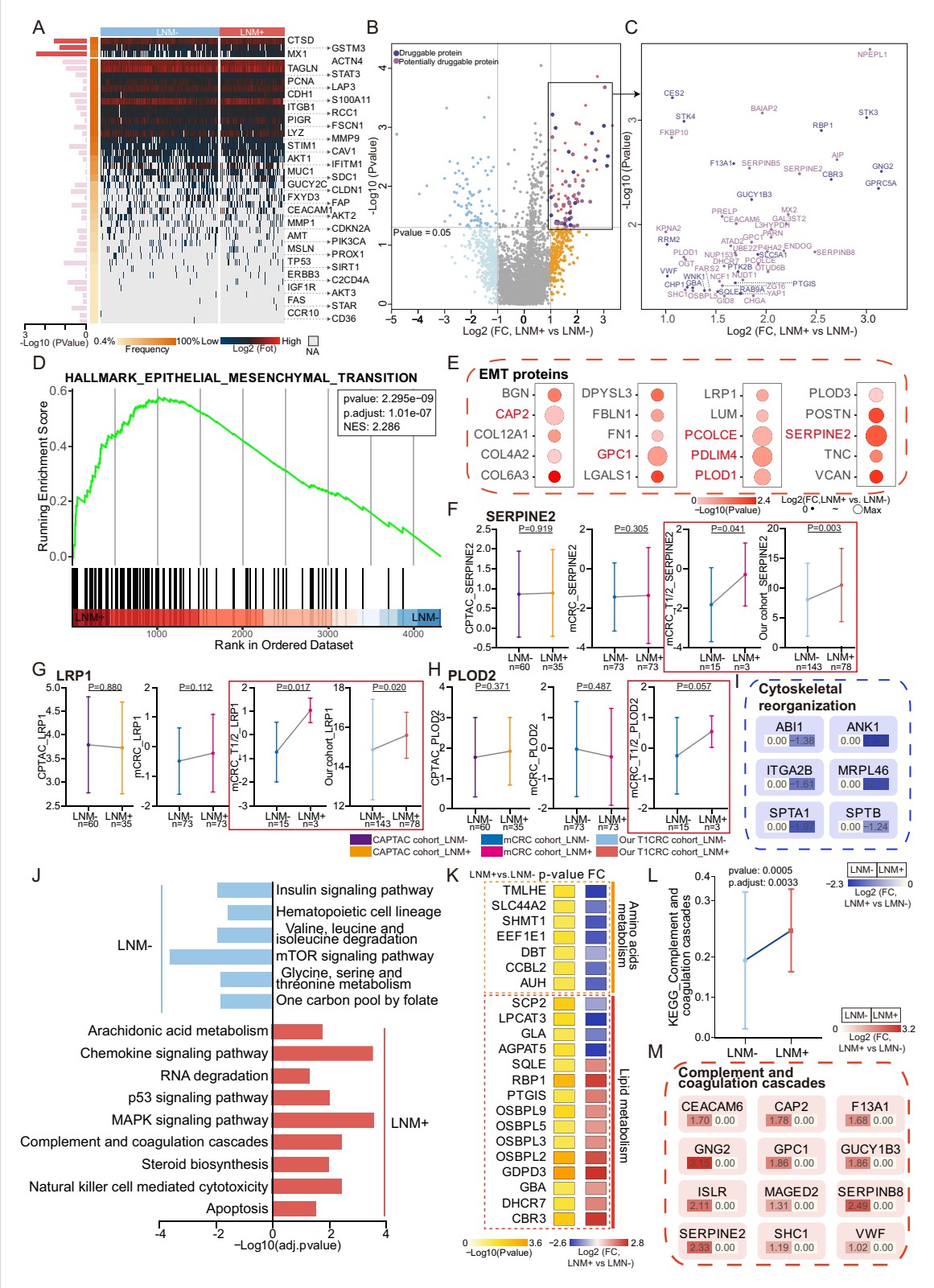

**Figure 2.** Protein signatures and functional differences between lymph node metastasis (LNM)-negative and LNM-positive patients with T1 colorectal cancer (CRC). (**A**) Forty-four reported protein markers associated with LNM in CRC. (**B**) A volcano plot showing proteins overexpressed in LNM-negative or LNM-positive patients (light blue and orange indicate proteins found in >30% of samples and a fold change of more than two, whereas blue and red indicate proteins with p<0.05; other proteins are shown in gray). Dark and light purple represent druggable and potentially druggable proteins based

*Figure 2 continued*

on the Drug Gene Interaction Database (http://www.dgidb.org/). (**C**) A scatterplot showing druggable (dark purple, N=19) and potentially druggable (light purple, N=34) proteins based on the Drug Gene Interaction Database (http://www.dgidb.org/) overexpressed in LNM-positive patients. (**D**) Gene set enrichment analysis plot of the Hallmark Epithelial Mesenchymal Transition (EMT) gene set, identified as significantly enriched (FDR of <0.05) using unbiased geneset enrichment analysis. (**E**) Details of proteins involved in the EMT. (**F, G, H**) Comparison of SERPINE2 (**F**), LRP1 (**G**), and PLOD2 (**H**) expression between LNM-negative group and LNM-positive group in CPTAC cohort, mCRC cohort, T1/2 patients of mCRC cohort and our cohort (Wilcoxon rank-sum test). (**I**) Details of proteins involved in the cytoskeletal remodeling. (**J**) Single sample Gene Set Enrichment Analysis (ssGSEA) of LNM-negative group patients compared with LNM-positive group patients. (**K**) Dysregulation of metabolic bioprocesses in T1 CRC. Alterations of representative proteins depicted as-log10 p-value and log2 FC (LNM+/LNM−, Wilcoxon rank-sum test). (**L & M**) Comparison of Complement and coagulation cascade scores between the LNM-negative group and LNM-positive (LNM+/LNM−, limma approach) (**L**) and the details of proteins involved in the cytoskeletal remodeling and coagulation cascades (**M**).

The online version of this article includes the following source data and figure supplement(s) for figure 2:

**Source data 1.** Log2 transformed proteomics data.

**Source data 2.** Druggability based on the Drug Gene Interaction Database.

**Source data 3.** GSEA.

**Source data 4.** Immune composition of T1 colorectal cancer (CRC) from xCell.

**Source data 5.** D2-40 immunohistochemistry (IHC) staining score.

**Figure supplement 1.** Protein signatures across three cohorts.

**Figure supplement 2.** Protein differences by differentiation and histologic type and proteogenomic characteristics of mucinous colorectal adenocarcinoma.

research has found that loss of ABI1 contributes to tumor progression through regulation of the EMT-WNT pathway. Cytoskeletal remodeling-related pathway (GO: 0003774) was also found to be up-regulated in LNM-negative patients of the mCRC cohort (*Figure 2—figure supplement 1D*). The rearrangement of cytoskeletal proteins might also be responsible for LNM in patients with T1 CRC though EMT (*Figure 2I*).

We employed single sample Gene Set Enrichment Analysis (ssGSEA) in 166 significantly different expressed proteins (log2FC >1 or <-1 and p<0.05, Wilcoxon rank-sum test) (identified in at least 30%, log2FC >1 or <−1 and p<0.05, Wilcoxon rank-sum test) (*Figure 2J*). Amino acid metabolism pathways, such as 'valine, leucine, and isoleucine degradation' (hsa00280) and 'glycine, serine, and threonine metabolism' (hsa00260), were found to be enriched in the LNM-negative group (adjusted p≤0.05, limma approach), and related proteins (TMLHE, SLC44A2, SHMT1, and EEF1E1) were up-regulated in the LNM-negative group (Log2FC <1, p<0.05) (*Figure 2K*). On the other hand, the LNM-positive group mainly expressed lipid metabolism pathways, such as 'arachidonic acid metabolism' (hsa00590) and 'steroid biosynthesis' (hsa00100), which are known to promote cancer cell proliferation and migration and are involved in the regulation of EMT. In addition, tumor metastasis-related signaling pathways, such as the MAPK and p53 pathways, as well as cellular process categories (NK cell-mediated cytotoxicity and apoptosis), were enriched in LNM-positive patients. Meanwhile, the mTOR signaling pathway was enriched in the LNM-negative group. The MSI-related protein, MLH1, was significantly up-regulated in LNM-positive group (*Figure 2—figure supplement 1E*). We also performed comparative pathway enrichment analysis of the differential expressed proteins (LNM + vs. LNM−: p-value <0.05, Wilcoxon rank-sum tests) under different observation percentiles (more than 10%, 30%, and 50%). The results show that the main change pathways are similar at different observation percentiles (*Figure 2—figure supplement 1F and G*). Overall, these findings suggest that different metabolic pathways and signaling pathways might contribute to LNM in T1 CRC.

Interestingly, the LNM-positive group was significantly enriched in coagulation cascades (*Figure 2L*), which occurred during EMT, and previous studies have indicated that the inhibition of coagulation greatly limits cancer metastasis (*Gil-Bernabé et al., 2013*). Twelve of the 84 proteins that were elevated in LNM-positive patients (log2FC >1 and p<0.05, Wilcoxon rank-sum test) were related to coagulation cascades (*Figure 2M*), and many of these 12 proteins are known or suspected to be linked to CRC or other cancer metastasis. For instance, CEACAM6 and SERPINE2 are risk genes for colorectal liver metastases (*Beauchemin and Arabzadeh, 2013*; *Bergeron et al., 2010*). VWF, SHC1, and CAP2 have been reported to be elevated in LNM-positive patients with gastric cancer (*Franchini et al., 2013*; *Liebermeister et al., 2014*; *Li et al., 2020a*). Moreover, F13A1 and GPC1 are

biomarkers for melanoma metastasis (*Azimi et al., 2014*). Meanwhile, we found the negative correlations between the cytoskeletal remodeling-related proteins and coagulation cascades, for example, the expression of SPTA1 was negatively correlated with seven coagulation cascades-related proteins including CEACAM6, MAGED2, ISLR, GPC1, SHC1, SERPINB8 and GNG2, MRPL46 showed a negative correlation with five coagulation cascades related proteins. ANK1 and SPTB were also found to be negatively correlated with coagulation cascades. Recent studies have shown that cytoskeletal remodeling can affect complement and coagulation pathway activation. SPTA1 and SPTB are components of the erythrocyte cytoskeleton, and defects in these proteins can lead to hemolytic anemia, which can activate the coagulation cascade. MRPL46 is a mitochondrial ribosomal protein that may play a role in mitochondrial function, which can affect coagulation and inflammation (*Figure 2—figure supplement 1H*).

In conclusion, unique protein markers specific to T1 CRC were found in this study, that may improve the understanding of LNM in early-stage CRC. EMT pathway was significantly enriched in the LNM-positive group of T1/2 CRC, and EMT-related proteins were found to be specifically up-regulated in the early-stage CRC LNM-positive group compared to the LNM-negative group. We also found differences in metabolic pathways and signaling pathways between the LNM-positive and negative groups, suggesting that different metabolic pathways and signaling pathways may contribute to LNM in T1 CRC.

## Characterization of mucinous colorectal adenocarcinoma

In agreement with previous reports (*Ikematsu et al., 2013*), in our T1 CRC cohort the ratio of LNM was significantly higher in patients with poorly poor differentiated patients (48.6%) compared with well-moderately differentiated patients, in T1 CRC (28.8%) (p=0.004, chi-square test), Mucinous adenocarcinoma patients (55%) also shown a greater tendency to LNM than adenocarcinoma (30.1%) (p=0.004, chi-square test) (*Figure 1—figure supplement 1*, *Figure 2—figure supplement 2A*), and mucinous adenocarcinoma was considered to be a significant risk factor of LNM in T1 CRC (*Xu et al., 2020*). More than 50% of patients with mucinous colorectal adenocarcinoma had LNM in our study. Thus, we divided our cohort into three subgroups: those with well to moderately differentiated adenocarcinoma (DS1, N=149; LNM ratio: 43 of 149), poorly differentiated adenocarcinoma (DS2, N=32; LNM ratio: 13 of 32) and mucinous adenocarcinoma (DS3, N=40; LNM ratio: 22 of 40) (p=0.007, chi-square test) (*Figure 2—figure supplement 2A* and *Figure 2—source data 1*). To identify the molecular characteristics of different groups, we determined the significantly changed proteins in each group (identified in at least 30% and p<0.05, Kruskal-Walli's test across groups with a log2-fold change [log2FC]>1, mean of DS1/DS2/DS3 vs mean of other two groups) (*Figure 2—figure supplement 2B* and *Figure 1—figure supplement 2H*). As a result, 140 proteins were overexpressed in DS1(log2FC >1, mean of DS1 vs mean of DS2 and DS3; p<0.05, Kruskal-Walli's test). Further pathway enrichment analysis revealed that the oxidative phosphorylation and TCA cycle pathways were higher in DS1 than in DS2 and DS3 (p<0.05) (*Figure 2—figure supplement 2C*). In DS2, 178 proteins were highly expressed (log2FC >1, mean of DS2 vs mean of DS1 and DS3; p<0.05, Kruskal-Walli's test) and enriched in the GTPase activity and Wnt signaling pathways. In DS3, 326 proteins were overexpressed (log2FC >1, mean of DS3 vs mean of DS1 and DS2; p<0.05, Kruskal-Walli's test), functioning in more aggressive pathways such as ECM organization, cell migration, and vesicle-mediated transport. We also found that DS2 shared some features with the other two groups. For example, a SUMOylation-related pathway was identified in both DS1 and DS2, and both DS2 and DS3 were characterized by elevated levels of the NFkB signaling pathway. In conclusion, oxidative phosphorylation and TCA cycle pathways were enriched in the well to moderately differentiated adenocarcinoma subgroup, while GTPase activity and Wnt signaling pathways were enriched in poorly differentiated adenocarcinoma. The mucinous adenocarcinoma subgroup was characterized by aggressive pathways such as ECM organization, cell migration, and vesicle-mediated transport.

We then studied the relationship between the overrepresented proteins according to histological type and LNM-related proteins. Interestingly, we found a significant negative correlation in the mucinous adenocarcinoma group (Pearson correlation coefficient = −0.53, p<0.05) but not in the other two groups, indicating that mucinous adenocarcinoma has a unique LNM mechanism (*Figure 2—figure supplement 2D*). Mucinous adenocarcinoma of the colorectal is a lethal cancer with unknown molecular etiology and a high propensity to lymph node metastasis. Previous proteomic studies on

mucinous adenocarcinoma have found the proteins associated with treatment response in rectal mucinous adenocarcinoma and mechanisms of metastases in mucinous salivary adenocarcinoma (*Sun et al., 2022*; *Panaccione et al., 2017*). Mucinous adenocarcinoma is a distinct subtype of adenocarcinoma and is characterized by abundant mucinous components. In our data, glycoproteins and related enzymes were overexpressed in the mucinous adenocarcinoma group compared with the nonmucinous adenocarcinoma group (*Figure 2—figure supplement 2E*), indicating that glycoproteins are an important component of mucinous adenocarcinoma.

To explore the mechanism of LNM in mucinous adenocarcinoma in T1 CRC, we focused on the functions and characterizations of the 326 proteins that were highly expressed in the mucinous adenocarcinoma group. We found that six integrins (ITGA1, ITGAV, ITGA11, ITGA9, ITGB3, and ITGB5) and 11 reported extracellular vesicle (EV) markers (ADIRF, CSPG4, DPP4, ENTPD1, LRG1, PECAM1, PLVAP, RAB25, TMEM2, TTR, and YBX1) were overexpressed in the mucinous adenocarcinoma group (*Figure 2—figure supplement 2F, G*). Furthermore, proteins involved in the membrane trafficking and/or vesicle transport pathways, such as ARFGAP1, ARRDC1, and TOR1A. and proteins involved in ECM organization, including DDR1, LAMA5, and TTR, were upregulated in mucinous colorectal adenocarcinoma (*Figure 2—figure supplement 2H*). Next, the composition of the tumor microenvironment in our cohort was studied using xCell (*Aran et al., 2017*; *Figure 2—figure supplement 2I* and *Figure 2—source data 4*). The stromal score was significantly higher in mucinous colorectal adenocarcinoma than in non-mucinous adenocarcinoma (p<0.05, Kruskal-Wallis test), and the signatures of endothelial cells, smooth muscle cells, and osteoblasts were enriched in mucinous colorectal adenocarcinoma; however, pericytes were decreased. The increase in endothelial cells, especially lymphatic endothelial cells, and the decrease in pericytes indicated that in mucinous colorectal adenocarcinoma, intratumoral lymphatic vessels rather than blood vessels might increase. Blood vessels are composed of endothelial cells and pericytes (*dela Paz and D'Amore, 2009*), and lymphatic vessels are composed of a monolayer of endothelial cells (*Baluk et al., 2007*). This suggests that in mucinous colorectal adenocarcinoma, the density of intratumoral lymphatic vessels, rather than blood vessels, is increased, and an increase in intratumoral lymphatic vessel density might correlate with LNM (*Lin et al., 2010*). We employed immunohistochemistry (IHC) to validate observation using commercially available D2-40 (PDPN), a lymphatic endothelial marker, antibodies. Previous studies have shown that PDPN expression correlated with LNM in numerous cancers, especially in early oral squamous cell carcinomas (*Huber et al., 2011*). Immunostaining showed the D2-40 expression in mucinous adenocarcinoma was significantly higher than that in non-mucinous adenocarcinoma, in agreement with the proteomics data (*Figure 2—figure supplement 2J* and *Figure 2—source data 5*). In summary, we hypothesize that mucinous adenocarcinoma impacts the tumor microenvironment through EVs, resulting in an increase in intratumoral lymphatic vessel density, thereby promoting LNM (*Figure 2—figure supplement 2K*).

## Discriminative classifier to identify T1 CRC with LNM

To discover protein markers that can be used to predict LNM in patients with T1 CRC, we established a classifier that could effectively distinguish LNM to guide clinical decisions for T1 CRC. Among 70 LNM-negative and 62 LNM-positive patients in the training cohort, we identified 407 candidate proteins (identified in at least 30% of samples and p<0.1, Wilcoxon rank-sum test in the training cohort) (*Figure 1—figure supplement 2H*). To determine feature importance (significance of the prediction feature), we employed least absolute shrinkage and selection operator (LASSO) logistic regression and used the LNM status (negative or positive) to determine the discrimination power of each signature (*Figure 3—figure supplement 1A* and B). We constructed a classifier using the intensities of 55 proteins that facilitated accurate discrimination between LNM-negative and LNM-positive patients with T1 CRC in the training cohort (*Figure 3A*, *Figure 1—source data 3* and *Figure 3—source data 1*). The classifier achieved an AUC of 1.00 (95% CI, 1.000) through 10-fold cross-validation in the training cohort, which indicated higher predictive power for LNM than the NCCN guidelines (AUC = 0.561) (p<0.001) (*Figure 3B*). The Youden's index-derived cutoff was used as the threshold, and the 55-protein classifier yielded 100% sensitivity and specificity, whereas the NCCN guidelines yielded 93.5% sensitivity and 18.6% specificity. In summary, the 55-protein classifier could be used to more accurately predict the individual probability of LNM in the training cohort (*Figure 3B*).

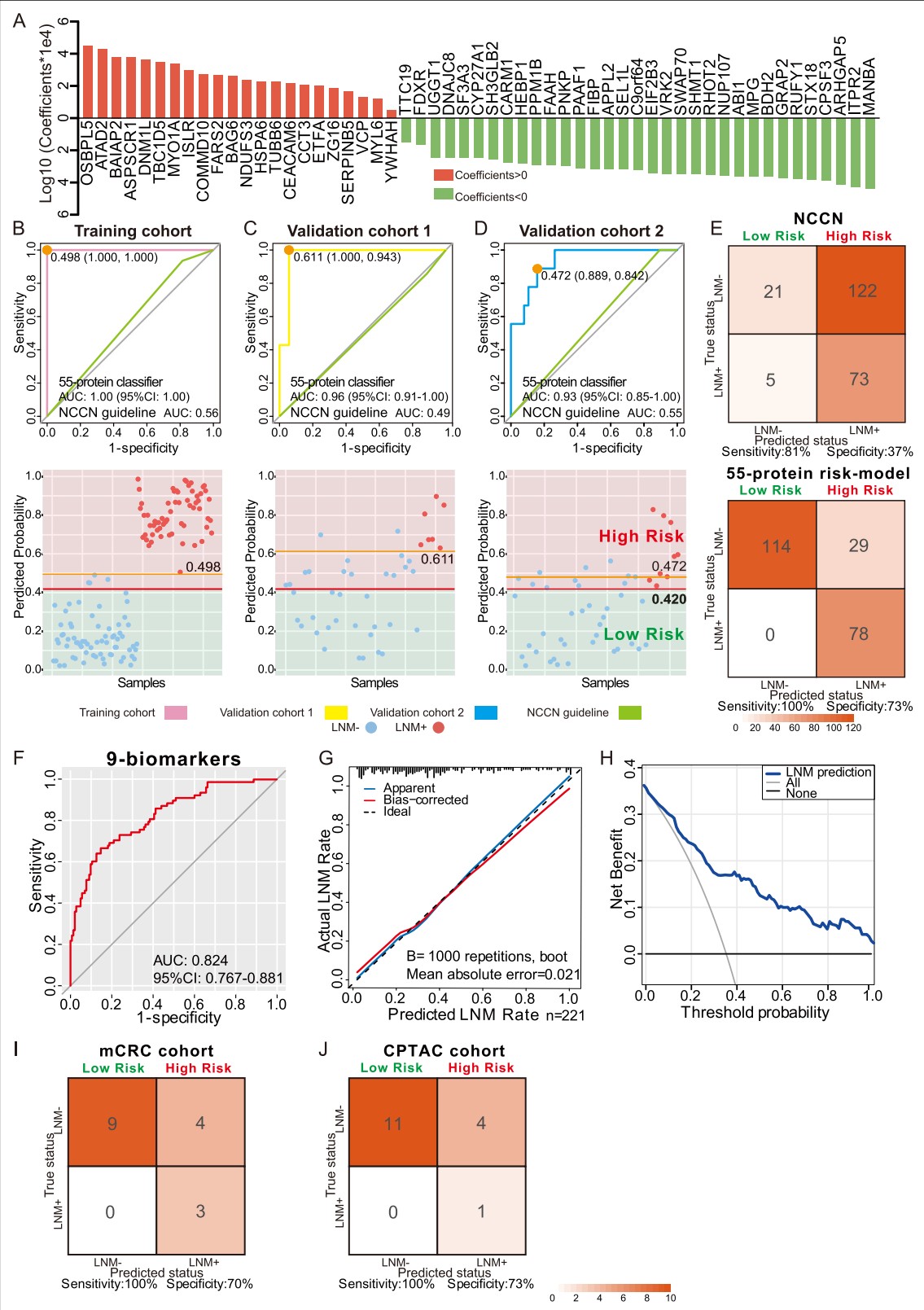

**Figure 3.** Development and validation of a protein classifier to predict lymph node metastasis (LNM) with T1 colorectal cancer (CRC). (**A**) The predictive relevance of all 55 protein markers to distinguish LNM-positive from LNM-negative T1 CRC patients is represented by a bar chart, and their least absolute shrinkage and selection operator (LASSO) coefficients are indicated. Also, see *Figure 1—figure supplement 2H*, *Figure 3—figure supplement 1A, B*. (**B, C, D**) Top: Receiver operating characteristic (ROC) curve with the area under the curve (AUC) for the protein classifier of the

*Figure 3 continued on next page*

*Figure 3 continued*

training cohort (**B**), validation cohort 1 (**C**), and validation cohort 2 (**D**). Bottom: Scatterplot representing the score of each patient with (red dot) or without (blue dot) LNM, the optimal threshold (Youden's index) of each curve (orange line) and the safety cutoff line (red line). (**E**) Classification error matrix using NCCN guidelines and safety cutoff from our 55-protein model. (**F, G, H**) ROC curve of the optimized 9-biomarker classifier using binary logistic regression (**F**), calibration curve of the optimized model (**G**), and cost-benefit decision curves (**H**) in 221 patients. (**I & J**) Classification confusion matrix of the simplified classifier in mCRC cohort (**I**) and CPTAC cohort (**J**). The number of samples identified is noted in each box.

The online version of this article includes the following source data and figure supplement(s) for figure 3:

**Source data 1.** Coefficients of 55 protein markers and the lymph node metastasis (LNM) scores of samples using least absolute shrinkage and selection operator (LASSO)-logistic regression.

**Source data 2.** Predicting risk score for lymph node metastasis (LNM) of each patient.

**Source data 3.** Coefficients of nine protein-markers.

**Source data 4.** External validation.

**Figure supplement 1.** Details of the least absolute shrinkage and selection operator (LASSO) regression model and immunohistochemistry (IHC) staining of targeted proteins.

To validate the predictive power of this classifier, a consecutive dataset from the ESD cohort (VC1, N=42) was adopted. All the samples in the ESD cohort (35 LNM-negative and seven LNM-positive individuals) were resected by endoscopy in the clinic. The 55-protein classifier achieved an AUC of 0.96 (95% CI, 0.917–1.000) in distinguishing LNM-positive from LNM-negative patients with 100% sensitivity/94.3% specificity in VC1 (*Figure 3C*). However, the AUC of the NCCN guidelines was 0.49 (85.7% sensitivity/11.4% specificity) (p<0.001) (*Figure 3C*). This result demonstrates that the proteomic classifier performs better than the NCCN guidelines.

To further assess the value of the proteomic classifier in the clinic, we utilized a prospective validation cohort (VC2) consisting of 47 patients (LNM-negative, N=38; LNM-positive, N=38; nine of whom received endoscopic resection, and the others received surgical resection. The classifier based on the intensities of the 55 proteins achieved 88.9% sensitivity and 84.2% specificity in distinguishing LNM-negative and LNM-positive T1 CRC patients, with an AUC of 0.93 (*Figure 3D*). For the NCCN guidelines, the AUC was 0.55 (p<0.001), which was relatively lower than that of the proteomic indicator.

To ensure the safety of those who have positive LNM, 0.420 was regarded as a cutoff value when stratifying the patients into 'high-risk' and 'low-risk' groups by predicting risk score for LNM range from 0 to 1, and at this threshold, all patients in the low-risk group were LNM-negative (*Figure 3B–D* and *Figure 3—source data 2*). When we used current clinical treatment guidelines (NCCN guidelines), it resulted in stratifying 88% patients (195 of the 221) into a high-risk category and the remaining 12% (26 of 195) into a low-risk group, only 37% (73 of 195) of patients are actually high risk, 63% (122 of 195) of patients underwent unnecessary additional surgery, and there are five (9.2%, 5 of 26) LNM-positive cases mistakenly assigned to the low-risk group (*Figure 3E* and *Figure 1—source data 1*). In contrast, of the 107 patients who are classified as high risk by our model, 78 had LNM (72%), indicating that only 27.1% (29 of 107) of all patients with T1 CRC were overtreated, and all the patients stratified into the low-risk group are LNM-negative (*Figure 3E* and *Figure 3—source data 2*).

The cutoff of p-value less than 0.05 or identification frequency more than 50% was also used to screen for variables, and 355 or 323 protein markers were identified. The lasso regression was carried out, and with AUC values of 1.000, 0.824, and 0.918 for the training cohort, VC1, and VC2 under the cutoff of p-value less than 0.05; AUC values of 0.999, 0.812, and 0.886 under the cutoff of identification frequency more than 50%, respectively (*Figure 3—figure supplement 1C*). The results suggested that proteomic/protein expression profile could be used to distinguish the risk of LNM in T1 CRC.

In conclusion, compared to current strategies, our 55-protein classifier can more accurately distinguish LNM-negative and LNM-positive T1 CRC patients to better guide clinical decision-making and determine whether a patient needs additional surgery after ESD.

## Simplified classifier to Identify T1 CRC with LNM

To further reduce the complexity of the indicator, 19 proteins with log2-fold change [log2FC]>1 and p<0.05 (Wilcoxon rank-sum test) in 221 samples were selected from 55 protein markers (SHMT1, PAAF1, VRK2, SEL1L, ITPR2, CPSF3, ABI1, RHOT2, SWAP70, and TTC19 were expressed higher in 143 LNM-negative patients, whereas OSBPL5, FARS2, ZG16, ATAD2, CEACAM6, SERPINB5,

COMMD10, BAIAP2, and ISLR were expressed higher in 78 LNM-positive patients), followed by multiple logistic regression analysis that resulted in a final model comprising nine proteins (ATAD2, CEACAM6, COMMD10, FARS2, ITPR2, RHOT2, SERPINB5, SWAP70, VRK2) (*Figure 3—source data 3*). The 9-protein classifier also demonstrated excellent performance in identifying LNM when we assessed its calibration, discrimination, and clinical usefulness (*Figure 3F–H*). After using bootstraps with 1000 resamples for validation, the AUC of the simplified model was 0.824 (95% CI, 0.767–0.881) (*Figure 3F*), and the calibration curve demonstrated good agreement between the predicted status and the true status, with an error of 0.021 (*Figure 3G*). The decision curve showed that when the threshold probability was >5%, the use of the 9-protein classifier to predict LNM added more benefit than either the treat-all-patients scheme or the treat-none scheme (*Figure 3H*).

## External validation of the simplified classifier

To evaluate the generalizability and reliability of our model, the data from Bing Zhang's (mCRC cohort) and Chen Li's (CPTAC cohort) studies were used as external validation cohort, and our 9-protein simplified classifier was validated in these two data sets, respectively (*Zhang et al., 2014*; *Oh-e et al., 2001*).

First, we screened the data according to our inclusion criteria. Since the two external validation datasets focused on the all-stage CRC, the number of patients in the T1 stage was small, six patients for the mCRC cohort and three for the CPTAC cohort, we added T2NxM0 patients to the model validation.

We employed ComBat, an Empirical Bayes method, to reduce the batch effects between our dataset and the two datasets mCRC cohort and CPTAC cohort (*Figure 3—figure supplement 1E and F*). After batch correction, nine proteins from our simplified classifier were selected, and the prediction score was calculated for each sample from both the mCRC cohort and the CPTAC cohort, respectively (*Figure 3—source data 4* in revision). Consistent with our previous analyses, 0.420 was regarded as a cutoff value when stratifying the patients into 'high-risk' and 'low-risk' groups by predicting risk score for LNM range from 0 to 1, and at this threshold, all patients in the low-risk group were LNM-negative.

In mCRC cohort (T1/2 CRC patients, N=16; LNM−, N=13; LNM+, N=3), nine LNM-negative patients were classified into the low-risk group, and all the patients stratified into the low-risk group are LNM-negative, corresponding to a sensitivity of 100% and a specificity of 70% (*Figure 3I*).

In the CPTAC cohort (T1/2 CRC patients, N=16; LNM-, N=15; LNM+, N=1), 11 out of 15 LNM-negative patients were correctly identified, corresponding to a specificity of 73%. Although, the results were limited by the number of LNM-positive patients, LNM-positive patients were successfully assigned to the high-risk group with a sensitivity of 100% (*Figure 3J*).

These results indicated that our model was able to effectively identify patients with LNM during external validation, both in the mCRC cohort and CPTAC cohort, ensuring that no patients with metastasis were missed. At the same time, compared with the NCCN guidelines, our model classifies more patients without metastasis into the low-risk group, reducing the incidence of overtreatment, and would provide a valuable insight for clinical decisions to T1 CRC patients treatment.

## Five IHC biomarkers to predict LNM

To further validate the expression patterns of the biomarkers from the proteomic results, 13 proteins from the 19 differently expressed proteins were stained for in LNM-negative and LNM-positive FFPE specimens via IHC. IHC staining was first used in 22 FFPE T1 CRC cases for proteins higher expression in LNM-negative patients, and 21 FFPE T1 CRC cases for proteins higher expression in LNM-positive patients (*Figure 4—source data 1*). For the IHC analysis, protein abundance varied according to the adopted scoring system (on a scale of 0–12); thus, we generated a staining scale for all cases (*Figure 4A* and *Figure 4—figure supplement 1A*). In agreement with the proteomics data, staining of ABI1 (p=0.007), ITPR2 (p=0.054), and RHOT2 (p=0.011) showed an overall increase in LNM-negative patients (*Figure 4A and B*, Student's t-test), whereas ATAD2 (p=0.034) and ISLR (p=0.044) showed an overall increase in LNM-positive patients (*Figure 4A* and *Figure 4—figure supplement 1B*, Student's t-test) according to the proteomics data. However, the protein levels of BAIAP2, CEACAM6, PAAF1, SHMT1, SWAP70, TTC19, VRK2, and ZG16 were not significantly different between LNM-negative and LNM-positive patients (*Figure 4—figure supplement 1A*).

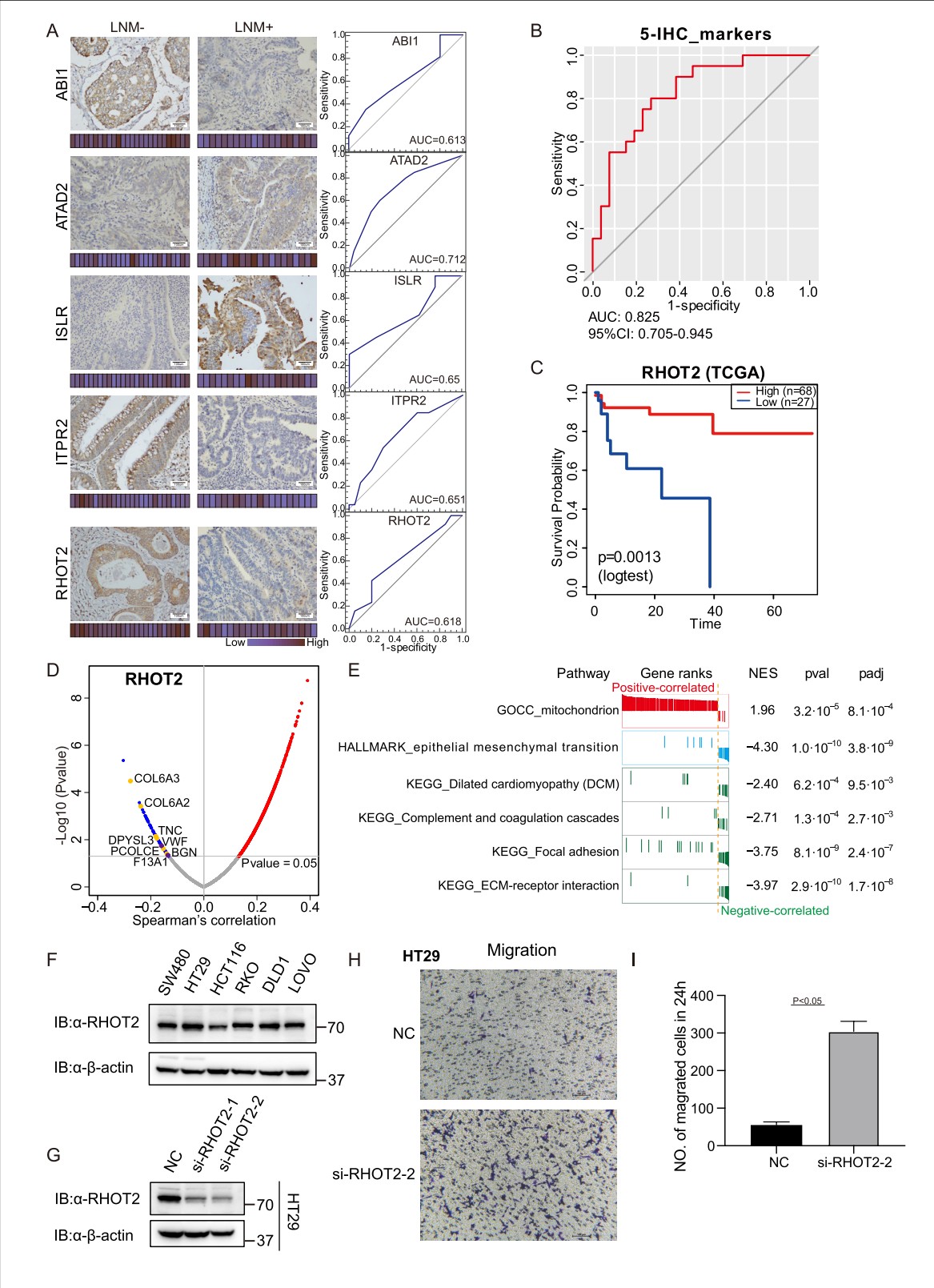

**Figure 4.** Immunohistochemical staining of targeted proteins. (**A**) T1 colorectal cancer (CRC) samples from a set of 47 cases were used to verify the abundance of ABI1, ITPR2, RHOT2, ATAD2, and ISLR. The scores that represent the sum of the intensities and percentage of protein staining in the lymph node metastasis (LNM)-positive or LNM-negative patients are shown as a heat map. (Histological images were obtained using a × 40 objective, scale bars, 100um). The receiver operating characteristic (ROC) curve of each protein was built by their immunohistochemistry (IHC) score. (**B**) ROC curve

*Figure 4 continued on next page*

*Figure 4 continued*

of the five proteins classifier using IHC score by binary logistic regression. (**C**) The overall survival of patients with colon cancer was analyzed on the basis of The Cancer Genome Atlas (TCGA) database. (**D**) Correlations between RHOT2 activities and protein abundances in the T1 CRC cohort. (**E**) Gene set enrichment analysis (GSEA) pathways using the single-gene method of RHOT2. (**F**) The RHOT2 protein expression in human colon cancer cells (SW480, HT29, HCT-116, RKO, DLD1, and LoVo) was measured by western blotting. (**G**) The protein expression of RHOT2 in HT29. (**H & I**) The migration ability of HT29 cells was detected by transwell assay (images were obtained using a × 20 objective, scale bars, 100um) (Student's t-test).

The online version of this article includes the following source data and figure supplement(s) for figure 4:

**Source data 1.** Immunohistochemistry (IHC) staining score.

**Source data 2.** RHOT2 western blot source images.

**Figure supplement 1.** Immunohistochemical staining of targeted proteins.

To further validate the immunohistochemical results, more patients were added. We examined the expression of ABI1, ITPR2, RHOT2, ATAD2, and ISLR in a total of 47 T1 CRC cases (27 LNM-negative patients and 20 in LNM-positive patients). We then evaluated the predictive power of individual proteins to distinguish patients with and without LNM by the AUC. All five differentially expressed proteins, namely, ABI1 (AUC = 0.613), ITPR2 (AUC = 0.651), RHOT2 (AUC = 0.618), ATAD2 (AUC = 0.712), and ISLR (AUC = 0.65), showed good discrimination according to their IHC scores (*Figure 4A*). Using binary logistic regression to analyze the results of these proteins, we obtained a 5-protein classifier using the IHC score. The 5-protein IHC classifier achieved an AUC of 0.825 in 47 patients (*Figure 4B*).

## RHOT2 promotes the migration and invasion of colon cancer cells

It has previously been reported that low levels of ABI1 and high levels of ADAT2 or ISLR, result in an increase in extracellular matrix (ECM) degradation, migration, and cell invasion in colon cancer. However, whether RHOT2 could affect the LNM in colon cancer remains unclear. RHOT2 was significantly down-regulated (Log2FC = −1.35; p=0.003, Wilcoxon rank-sum test) in LNM-positive patients compared with LNM-negative patients in our T1 CRC cohort. Furthermore, as shown in *Figure 4C*, the analysis on the basis of the TCGA database suggested that the low level of RHOT2 is related to the low overall survival of patients with colon cancer (*Figure 4C*, p<0.05) (*Zhang et al., 2014*).

To further ascertain the function of RHOT2 in T1 CRC, the correlations between the expression of RHOT2 and other proteins were calculated in our cohort (*Figure 4D*). We found 1508 proteins were correlated significantly (p<0.05, Spearman) with RHOT2, and 1354 proteins showed a positive correlation (coefficient >0) with RHOT2, 154 proteins were negatively correlated with RHOT2 (coefficient <0) (*Figure 4D*). However, when we performed GSEA in RHOT2-associated proteins to identify biological signatures impacted by RHOT2, most of the obtained pathways (p<0.01) showed NES less than 0, which means these pathways were mainly enriched in the RHOT2-negative-correlated group, only 'mitochondrion' (GOCC) had a positive correlation. As we know RHOT2 is an important protein involved in the regulation of mitochondrial dynamics and mitophagy (*Fransson et al., 2006*). This result indicates that the involvement of RHOT2 in regulation of mitochondrial function might contribute to the pathogenesis of metastasis in cancer, especially in early-stage CRC (*Figure 4E*). As expected, the RHOT2-negative-correlated group was significantly enriched in EMT (HALLMARK) and complement and coagulation cascade pathways. Proteins up-regulated in LNM-positive group (LNM +vs. LNM-: Log2FC >0; p<0.05, Wilcoxon rank-sum test) were negatively correlated with RHOT2 (p<0.05, coefficient <0, Spearman), including CAP2, COL6A3, COL6A2, TNC, DPYSL3, PCOLCE, and BGN in pathway EMT; and GUCY1B3, VWF, and F13A1 in pathway complement and coagulation cascades (*Figure 2E and L*; *Figure 4D*). ECM, focal adhesion, and Dilated cardiomyopathy (DCM) pathways were also enriched in the negative-correlated group (*Figure 4E*). Degradation of RHOT2 has already been reported to be associated with DCM (*Cao et al., 2019*). Overall, RHOT2 might play an important role in T1 CRC LNM.

Then, we detected the RHOT2 expression in human colon cancer cells (SW480, HT29, HCT-116, RKO, DLD1, and LoVo) by western blot and RHOT2 was confirmed to be expressed in all colon cancer cell lines (*Figure 4F* and *Figure 4—source data 2*). Cells were confirmed by STR profiling and the negative results for mycoplasma contamination were ensured by PCR. To investigate the role of RHOT2 in the migration of colon cancer, RHOT2 interference fragments (si-RHOT2#1 and #2) were used in

this study. The data in *Figure 4G* and *Figure 4—source data 2* showed that the protein expression of RHOT2 was significantly decreased by si-RNAs, especially si-RHOT2#2. Then, we investigated the effect of RHOT2 in the migration of colon cancer cells. The results of the transwell assay showed that the migration ability of colon cancer cells was significantly increased by si-RHOT2 (*Figure 4H, I*, Student's t-test,p<0.05). Low expression level of RHOT2 markedly enhanced the migration ability of colon cancer cells (p<0.05).

## Discussion

Here, we present a comprehensive proteomics study to focus on LNM in patients with T1 CRCs. Using a mass spectrometry-based proteomic technique, we analyzed 221 T1 CRC patients and identified different molecular characteristics between patients with and without LNM. We also uncovered protein characteristics of mucinous colorectal adenocarcinoma.

Based on the T1 CRC proteomics dataset, we developed a high-performing protein signature-based model for predicting LNM, which can aid in making informed decisions regarding additional surgical resection following endoscopic resection of T1 CRC. The protein signatures successfully stratified patients according to their risk of LNM. In the training cohort, 105 candidate protein features were reduced to 55 potential predictors by examining the predictor-outcome association with the LASSO method. Based on these 55 biomarkers, 10-fold cross-validation was applied to the training cohort and generated a ROC curve. The classifier achieved an AUC of 1.00 in the training cohort, which was confirmed in two independent validation cohorts. The 55 protein markers achieved an AUC of 0.96 (95% CI, 0.917–1.000) in the ESD validation cohort and 0.933 (95% CI, 0.858–1.000) in the prospectively collected validation cohort. Besides, a simplified model consisting of only 9 proteins, which could effectively identify patients with LNM in T1/2 CRC, was developed and validated on external data.

In recent years, there have been many studies on LNM in T1 CRC, among which the following studies are representative: in 2021, Shin-Ei Kudo and colleagues developed a machine-learning artificial neural network using data on patients' age and sex, tumor characteristics, and histologic grade to identify patients with T1 CRCs who had LNM, the ANN model outperformed guidelines in identifying high-risk patients who require radical surgery. In 2022, Yinghui Zhao and colleagues conducted genome-wide methylation profiling of T1 CRC specimens and developed a nine-CpG signature that can distinguish LNM-positive versus LNM-negative specimens and pre-treatment biopsies. The signature has the potential to improve the selection of high-risk patients. Also in 2022, Yuma Wada and colleagues developed a blood-based liquid biopsy assay to detect LNM in high-risk submucosal T1 CRCs. Their transcriptomic panel of four miRNAs and five mRNAs showed robust identification of patients with LNM.

Compared to the above-mentioned studies, in addition to higher predictive accuracy, this study has the following advantages: since the ultimate goal of the project was to precisely identify the LNM in T1 CRC, and to reduce additional surgery after endoscopic resection for LNM-negative patients, we set up a cohort only containing EDS samples to validate our predictive model, and a single-blinded prospective cohort was also used to validate the accuracy of the model. Furthermore, we demonstrated the potential for clinical translation of our study. Based on the proteomic results, we further constructed an effective classifier using five proteins based on their IHC score to optimize our model for clinical use.

In this study, to ensure the accuracy of the LN status of the enrolled patients, the dissected number of LN in all patients including both surgical resection and ESD was more than 12. However, the longer-term follow-up data, including DFS, PFS, etc., are not available due to limitations in sample collection time and the prognosis of such patients needs to be tracked over long periods of time, and may impact the strength of our conclusions. To address this limitation, we used propensity-score matching to reduce confounding biases in our training cohort. Patients were prospectively enrolled in our validation cohort (VC2), which was designed as a single-blinded prospective study to enhance the rigor and reliability of our findings. Furthermore, the presence of isolated tumor cells (ITCs) or micrometastases (MMs) within regional LN is not considered, due to conventional histopathologic examination cannot detect them. According to previous studies, there were about 5% pT1N0 gastric cancer patients have ITCs in LN, and 10% in pT1Nx CRC (*Zhao et al., 2023*; *Yonemura et al., 2007*; *Ishida et al., 1997*; *Yasuda et al., 2007*; *Doekhie et al., 2006*). The effect of MMs on prognosis in pT1N0

CRC is still unclear. The present of ITCs/MMs in LN may explain why there are nearly 13% (29 of 221) of LNM-negative patients were classified into high-risk groups by the prediction model in our study. We will continuously follow the prognosis of the patients, and the ITCs/MMs in LN also need to be further validated in the future studies.

Besides, we found dysregulation of RHOT2 expression can impact LNM in T1 CRC. Although the function of RHOT2 in cancer is still unclear, the expression of its paralog RHOT1 affects metastasis in a variety of tumors, including pancreatic cancer, gastric cancer, small cell lung cancer, etc. (*Li et al., 2015*; *Peng et al., 2022*; *Zhang et al., 2021*). In addition, previous studies have found that Myc regulation of mitochondrial trafficking through RHOT1 and RHOT2 enables tumor cell motility and metastasis (*Agarwal et al., 2019*). In our research, we demonstrate that low expression of RHOT2 was associated with LNM and poor prognosis, and knocking down RHOT1 can significantly increase the migration of CRC cells. The mechanism by which RHOT2 affects CRC LNM is not fully understood, but might involve several pathways. One potential pathway involves the regulation of EMT, a process by which cancer cells acquire invasive properties and become more motile. More analytical studies and experiments are needed in our future research to understand the specific role and mechanism of RHOT2 in the process of tumor metastasis.

In conclusion, we used proteomic analysis to identify molecular characteristics and develop the high accuracy protein signature-based models for predicting LNM in patients with T1 CRC.

## Materials and methods

### Patient

In this study, a training cohort and two independent validation cohorts of consecutive patients who visited the General Surgery Department, Zhongshan Hospital, Fudan University (Shanghai, China) were enrolled. The study was approved by the Institution Review Board of Fudan University Zhongshan Hospital, approval number: B2019-166.

In the training cohort (a retrospective cohort), we initially collected data on 604 patients who underwent surgical resection with LN dissection between June 2008 and October 2019. We then excluded 233 patients in whom fewer than 12 LNs were examined from the LNM-negative group. After matching the two groups for sex and age by propensity-score, we enrolled 132 patients.

For the ESD validation cohort (VC1), we studied 42 consecutive ESD samples before additional surgical resection from June 2008 to October 2019, and the LNM status was then examined by LN dissection.

The prospective validation cohort (VC2) comprised 47 consecutively prospectively enrolled individuals from November 2019 to April 2020 and was a single-blinded cohort.

The model was developed and validated in the retrospective cohort and then prospectively tested in the prospective cohort. In all cohorts, the inclusion criteria (as follows) were the same: aged between 18 and 80 years; undergone curative surgical resection; and pathological confirmation of colorectal adenocarcinoma, mucinous adenocarcinoma, or signet-ring cell carcinoma with pT1 disease according to the AJCC/UICC TNM staging system, 8th edition. The exclusion criteria (as follows) were also the same in all cohorts: patients who had undergone only endoscopic treatment; those who were diagnosed with familial adenomatous polyposis, Lynch syndrome, or history of inflammatory bowel disease; those who had undergone transanal endoscopic microsurgery; those who developed synchronous invasive carcinomas; and those with missing data. Patients who had received preoperative chemotherapy or radiotherapy were excluded.

### Sample preparation

The FFPE samples derived from 221 T1 CRC patients were collected, and the tumor regions were determined by pathological examination. For clinical sample preparation, sections (10 μm thick) from FFPE blocks were macro-dissected, deparaffinized with xylene, and washed with ethanol. The ethanol was removed completely and the sections were left to air-dry. For this purpose, a hematoxylin-stained section of the same tumor was used as a reference. Areas containing 80% or more tumors were examined by pathologists.

Lysis buffer [0.1 M Tris-HCl (pH 8.0), 0.1 M DTT (Sigma, 43815), 1 mM PMSF (Amresco, M145)] was added to the extracted tissues before adding SDS to the solution. The solution with the samples

was sonicated for 1 min (3 s on and 3 s off, amplitude 25%) on ice. The supernatants were collected, and the protein concentration was determined using the Bradford assay. 4% sodium dodecyl sulfate (SDS) was added and kept for 2.5 hr at 99 °C with shaking at 1800 rpm. The solution was collected by centrifugation at 12,000 × g for 5 min. A fourfold volume of acetone was added to the supernatant and kept at –20 °C overnight. Subsequently, the acetone-precipitated proteins were washed three times with cooled acetone. Filter-aided sample preparation (FASP) procedure was used for protein digestion (*Wiśniewski et al., 2009*). The proteins were resuspended in 200 µL 8 M urea (pH 8.0) and loaded in 30 kD Microcon filter tubes (Sartorius) and centrifuged at 12,000 g for 20 min. The precipitate in the filter was washed three times by adding 200 µL 50 mM NH4HCO3. The precipitate was resuspended in 50 µL 50 mM NH4HCO3. Protein samples underwent trypsin digestion (enzyme-to-substrate ratio of 1:50 at 37°C for 18–20 hr) in the filter, and then were collected by centrifugation at 12,000 g for 15 min. Additional washing, twice with 200 µL of MS water, was essential to obtain greater yields. Finally, the centrifugate was pumped out using the AQ model Vacuum concentrator (Eppendorf, Germany).

## Mass spectrometry analysis

Peptide samples were analyzed on a Q Exactive HF-X Hybrid Quadrupole-Orbitrap Mass Spectrometer (Thermo Fisher Scientific, Rockford, IL, USA) coupled with a high-performance liquid chromatography system (EASY nLC 1200, Thermo Fisher). Peptides, re-dissolved in Solvent A (0.1% formic acid in water), were loaded onto a 2 cm self-packed trap column (100 µm inner diameter, 3 µm ReproSil-Pur C18-AQ beads, Dr. Maisch GmbH) using Solvent A, and separated on a 150-µm-inner-diameter column with a length of 15 cm (1.9 µm ReproSil-Pur C18-AQ beads, Dr. Maisch GmbH) over a 75 min gradient (Solvent A: 0.1% formic acid in water; Solvent B: 0.1% formic acid in 80% ACN) at a constant flow rate of 600 nL/min (0–75 min, 0 min, 4% B; 0–10 min, 4–15% B; 10–60 min, 15–30% B; 60–69 min, 30–50% B; 69–70 min, 50–100% B; 70–75 min, 100% B). The eluted peptides were ionized under 2 kV and introduced into the mass spectrometer. MS was operated under a data-dependent acquisition mode. For the MS1 Spectra full scan, ions with m/z ranging from 300 to 1400 were acquired by Orbitrap mass analyzer at a high resolution of 120,000. The automatic gain control (AGC) target value was set as 3E+06. The maximal ion injection time was 80ms. MS2 Spectra acquisition was performed in top-speed mode. Precursor ions were selected and fragmented with higher energy collision dissociation with a normalized collision energy of 27%. Fragment ions were analyzed using an ion trap mass analyzer with an AGC target value of 5E+04, with a maximal ion injection time of 20 ms. Peptides that triggered MS/MS scans were dynamically excluded from further MS/MS scans for 12 s. A single-run measurement was kept for 75 min. All data were acquired using Xcalibur software (Thermo Scientific).

## Peptide and protein identification

MS raw files were processed using the Firmiana proteomics workstation (*Feng et al., 2017*). Briefly, raw files were searched against the NCBI human Refseq protein database (released on 04-07-2013; 32,015 entries) using the Mascot search engine (version 2.3, Matrix Science Inc). The mass tolerances were: 20 ppm for precursor and 50 mmu for product ions collected by Q Exactive HF-X. Up to two missed cleavages were allowed. The database searching considered cysteine carbamidomethylation as a fixed modification, and N-acetylation, and oxidation of methionine as variable modifications. Precursor ion score charges were limited to +2, +3, and +4. For the quality control of protein identification, the target-decoy-based strategy was applied to confirm the FDR of both peptide and protein, which was lower than 1%. Percolator was used to obtain the quality value (q-value), validating the FDR (measured by the decoy hits) of every peptide-spectrum match (PSM), which was lower than 1%. Subsequently, all the peptides shorter than seven amino acids were removed. The cutoff ion score for peptide identification was 20. All the PSMs in all fractions were combined to comply with a stringent protein quality control strategy. We employed the parsimony principle and dynamically increased the q-values of both target and decoy peptide sequences until the corresponding protein FDR was less than 1%. Finally, to reduce the false positive rate, the proteins with at least one unique peptide and more than $10^{-5}$ in FOT were selected for further investigation. Keratins were also excluded to ensure the credibility of the data.

## Label-free-based MS quantification of proteins

The one-stop proteomic cloud platform 'Firmiana' was further employed for protein quantification. Identification results and the raw data from the mzXML file were loaded. Then for each identified peptide, the extracted-ion chromatogram (XIC) was extracted by searching against the MS1 based on its identification information, and the abundance was estimated by calculating the area under the extracted XIC curve. For protein abundance calculation, the nonredundant peptide list was used to assemble proteins following the parsimony principle. The protein abundance was estimated using a traditional label-free, intensity-based absolute quantification (iBAQ) algorithm (*Schwanhäusser et al., 2011*), which divided the protein abundance (derived from identified peptides' intensities) by the number of theoretically observable peptides. We built a dynamic regression function based on the commonly identified peptides in tumor samples. According to correlation value R2, Firmiana chose linear or quadratic functions for regression to calculate the retention time (RT) of corresponding hidden peptides, and to check the existence of the XIC based on the m/z and calculated RT. Subsequently, the fraction of total (FOT), a relative quantification value was defined as a protein's iBAQ divided by the total iBAQ of all identified proteins in one experiment, and was calculated as the normalized abundance of a particular protein among experiments. Finally, the FOT was further multiplied by $10^5$ for ease of presentation, and FOTs less than $10^{-5}$ were replaced with $10^{-5}$ to adjust extremely small values (*Ge et al., 2018*).

## Immunohistochemistry (IHC) staining

FFPE tumor blocks were obtained from the Institute of Pathology at the Affiliated Zhongshan Hospital of Fudan University. Tumor blocks were cut into 4 µm sections. Nonspecific background staining was blocked via a serum-free protein blocker (BOSTER, USA) for 10 min at room temperature. Next, Sections were incubated with anti-SHMT1 (SignalwayAntibody, 31314), anti-PAAF1 (Proteintech, 17650–1-AP), anti-VRK2 (SignalwayAntibody, 43825), anti-ABI1 (SignalwayAntibody, 36723), anti-RHOT2 (Proteintech, 11237–1-AP), anti-SWAP70 (SignalwayAntibody, 42812), anti-TTC19 (Proteintech, 20875–1-AP), anti-ZG16 (Proteintech, 67389–1-Ig), anti-ATAD2 (Cell Signaling Technology, 78568 S), anti-BAIAP2 (Proteintech, 11087–2-AP), anti-ISLR2 (Novus Biologicals, AF4526-SP), anti-ITPR2 (SignalwayAntibody, 37666) overnight at 4 °C after blocking for 1 hr at room temperature, according to the manufacturers' instructions. Then, TMA sections were incubated with secondary biotinylated goat anti-Rabbit/Mouse antibody. For signal detection, samples were incubated with DAB (BOSTER, USA) for 10 min. All specimens were counterstained with hematoxylin and Scott's blue. Washing steps were conducted with tris-buffered saline with 0.1% Tween (pH 7.4). The IHC staining results were evaluated independently by two pathologists who were blinded to the clinicopathologic data. According to the proportion of positive cells, samples were scored as follows: 0+, none; 1+, <25%; 2+, 25–50%; 3+, 51–75%; and 4+, 75–100%. The staining intensity was evaluated as follows: 0, none; 1, weak; 2, medium; and 3, strong. The final score (range 0–12) was calculated by multiplying the two sub-scores, and divided into four groups: high (IHC score: 9–12), medium (IHC score: 5–8), low (IHC score: 1–4), and ND (IHC score: 0, not detected).

## MSI status evaluating

The Microsatellite instability (MSI) status of 171 patients was evaluated using IHC. Four mismatch repair (MMR) proteins (MLH1, MSH2, MSH6, and PMS2) were stained in tumor and normal samples, and the results were evaluated independently by two pathologists. The patients who showed positive staining of the nuclei of all four MMR proteins were considered as proficient MMR (pMMR)/ microsatellite stable (MSS), and the mismatch repair-deficient (dMMR) /microsatellite instability-high (MSI-H) cases show the loss of one of the two MLH1/PMS2 or MSH2/MSH6 heterodimers (*Samowitz, 2015*).

## Gene mutations

DNA of CRC FFPE tissues were extracted from serial sections (3 × 10 µm sections per extraction) using the QIAamp DNA FFPE Tissue Kit (Qiagen). according to the manufacturer's instructions. DNAs were quantitated by the PicoGreen Assay (Invitrogen, Carlsbad, CA). Analysis of mutations in the *KRAS*, *NRAS*, *BRAF*, and *PI3KCA* genes was carried out using four kits of the *KRAS/BRAF*, *NRAS*, *BRAF*, and *PI3KCA* Mutation Analysis Kit for Real-Time PCR (ABI 7500, applied biosystems Thermofisher). The

tests examined the most common mutations in codons 12, 13, 59, 61, 117, and 146 in *KRAS* and *NRAS* genes, in codon 600 of the *BRAF* gene and in codons 542, 545, 1047 in *PI3KCA* gene.

## Statistical analysis

Statistical details of experiments and analyses can be found in the figure legends and main text above. All statistical tests and calculations were performed using SPSS (SPSS Inc, Chicago, IL, USA), or R 3.5.1 (R Foundation for Statistical Computing, Vienna, Austria; http://www.r-project.org/). Protein intensities were log2-transformed for further analysis, apart from the coefficient analysis. Statistical significance tests, Chi-square test, and Wilcoxon rank-sum test, as denoted in each analysis. The statistical significance was considered when p-value <0.05. Kaplan–Meier plots (Log-rank test) were used to describe overall survival. The proteome analysis results were uniformly defined as 'up-regulated' or 'down-regulated' based on the measurements from a statistical test (two-sided Student's t-test/ Wilcoxon rank-signed test/ Kruskal-Walli's test) and the log2 transformed fold change (≥1 for upregulation and ≤ −1 for downregulation).

## LNM prediction model

The least absolute shrinkage and selection operator (LASSO) method, which is suitable for the regression of high-dimensional data was used to select the most useful predictive proteins from the training cohort. Lasso binary logistic regression was done using the 'glmnet' package. A LNM score was calculated for each patient via a linear combination of selected features that were weighted by their respective coefficients. To obtain an unbiased estimate of the prediction power of signatures, we performed a 10-fold cross-validation on the training cohort using Logistic Regression. The predictive power was also validated in two validation cohorts. Receiver operating characteristic (ROC) curves were constructed using the pROC R package in the R software, and the area under the ROC curve (AUC) was used to evaluate the diagnostic performance of the selected proteins.

Nine proteins with insignificant Estimate Std. Error z value (Pr (>|z|)<0.05) were further selected to build a simplified classifier using Logistic Regression. Calibration curves were plotted to assess the calibration of the classifier used rms and ResourceSelection R packages in the R software, accompanied by bootstrapping validation (1000 bootstrap resamples). Decision curve analysis was conducted to determine the clinical usefulness of the simplified model by quantifying the net benefits at different threshold probabilities in the validation dataset. The decision curve was constructed using the rmda R package in the R software.

The prediction power of IHC scores in single IHC samples was built by SPSS, and the five proteins predict model was built by R using Logistic Regression.

## Cell culture, transfection, and cell lines

We used human colon cancer cell lines SW480, HT29, HCT-116, RKO, DLD1, and LoVo (from General Surgery, Zhongshan Hospital) in this study. The cell lines were confirmed by STR profiling. The mycoplasma contamination of six cell lines was detected by the PCR-based test using GTB's Mycoplasma Detection Kit. None of the cells were contaminated with mycoplasma. All cell lines were cultured in DMEM (Corning Costar) with 10% FBS (Gibco), 100 units of penicillin, and 100 mg/mL streptomycin (Gibco) at 37 °C in 5% $CO_2$. Cells were transfected with siRNAs (si-RHOT2#1: sense (5'–3'), GCUCAACGCUUUCCAGAAATT; antisense (5'–3'), UUUCUGGAAAGCGUUGAGCTT; si-RHOT2#2: sense(5'–3'), GCGUCUACAAGCACCAUUATT; antisense (5'–3'), UAAUGGUGCUUGUAGACGCTT) using Lipofectamine 2000 according to the manufacturer's protocol. Knockdown efficiency was verified by western blotting.

## Cell migration analysis

For the transwell migration assay, $4 \times 10^4$ serum-starved cells were trypsinized and plated onto an FN-coated upper chamber membrane (8 µm pore filter, Corning Costar, Cat. No. 3422) of a transwell chamber with the corresponding inhibitors. The lower transwell chamber was filled with 2.5% serum containing DMEM. After incubation for 48 hr, the filters were removed, and the cells on the membrane were fixed with methanol. The migrated cells on the underside of the membrane were stained with 0.5% crystal violet. The dye was washed with water, and the cells were examined by microscopy.

## Ethics approval

The present study was carried out comply with the ethical standards of Helsinki Declaration II. The study was approved by the Institution Review Board of Fudan University Zhongshan Hospital, approval number: B2019-166. At the consent visit, patients were provided with a study overview. All patients signed the informed consent form and consent to publish. The informed consent document included consent for research samples and consent to protect confidential patient information by the personnel approved under the Institution Review Board of Fudan University Zhongshan Hospital. Any person not involved with the research study did not have access to patient identifying data. De-identified data was allowed to be shared with collaborators and findings from the study be published. Finally, the consent included patient's right to withdraw from the study at any time. The patient was provided with a copy of the signed informed consent.

## Acknowledgements

This work is supported by the National Key R&D Program of China (2022YFA1303200, 2022YFA1303201, 2020YFE0201600, 2018YFE0201600, 2018YFE0201603, 2018YFA0507500, 2018YFA0507501, 2017YFA0505100, 2017YFA0505102, 2017YFA0505101, 2017YFC0908404, 2016YFA0502500); sponsored by Program of Shanghai Academic/Technology Research Leader (22XD1420100); Shuguang Program of Shanghai Education Development Foundation and Shanghai Municipal Education Commission (19SG02); National Natural Science Foundation of China (31972933, 31770886, 31700682); the Major Project of Special Development Funds of Zhangjiang National Independent innovation Demonstration Zone ZJ2019-ZD-004; Shanghai Municipal Science and Technology Major Project (2017SHZDZX01); the Fudan original research personalized support project; CAMS Innovation Fund for Medical Sciences (CIFMS)[2019–12 M-5-063]; Shanghai Science and Technology Committee Project (19511121301); Clinical Research Plan of SHDC (SHDC2020CR5006).

## Additional information

### Funding

| Funder | Grant reference number | Author |
| --- | --- | --- |
| National Key Research and Development Program of China | 2022YFA1303200 | Chen Ding |
| Program of Shanghai Academic/Technology Research Leader | 22XD1420100 | Chen Ding |
| Shuguang Program of Shanghai Education Development Foundation and Shanghai Municipal Education Commission | 19SG02 | Chen Ding |
| National Natural Science Foundation of China | 31972933 | Chen Ding |
| Major Project of Special Development Funds of Zhangjiang National Independent Innovation Demonstration Zone | ZJ2019-ZD-004 | Chen Ding |
| Shanghai Municipal Science and Technology Major Project | 2017SHZDZX01 | Chen Ding |
| Fudan Original Research Personalized Support Project | | Chen Ding |

| Funder | Grant reference number | Author |
|---|---|---|
| Chinese Academy of Medical Sciences | Innovation Fund for Medical Sciences 2019-12M-5-063 | Chen Ding |
| Shanghai Science and Technology Committee | Project 19511121301 | Jianmin Xu |
| Clinical Research Plan of SHDC | SHDC2020CR5006 | Jianmin Xu |
| National Key Research and Development Program of China | 2022YFA1303201 | Chen Ding |
| National Key Research and Development Program of China | 2020YFE0201600 | Chen Ding |
| National Key Research and Development Program of China | 2018YFE0201600 | Chen Ding |
| National Key Research and Development Program of China | 2018YFE0201603 | Chen Ding |
| National Key Research and Development Program of China | 2018YFA0507500 | Chen Ding |
| National Key Research and Development Program of China | 2018YFA0507501 | Chen Ding |
| National Key Research and Development Program of China | 2017YFA0505100 | Chen Ding |
| National Key Research and Development Program of China | 2017YFA0505102 | Chen Ding |
| National Key Research and Development Program of China | 2017YFA0505101 | Chen Ding |
| National Key Research and Development Program of China | 2017YFC0908404 | Chen Ding |
| National Key Research and Development Program of China | 2016YFA0502500 | Chen Ding |
| National Natural Science Foundation of China | 31700682 | Chen Ding |
| National Natural Science Foundation of China | 31770886 | Chen Ding |

The funders had no role in study design, data collection and interpretation, or the decision to submit the work for publication.

## Author contributions

Aojia Zhuang, Conceptualization, Data curation, Software, Formal analysis, Investigation, Visualization, Methodology, Writing – original draft, Writing – review and editing; Aobo Zhuang, Conceptualization, Resources, Data curation, Software, Formal analysis, Investigation, Methodology, Writing – original draft; Yijiao Chen, Resources, Validation; Zhaoyu Qin, Formal analysis, Funding acquisition, Methodology, Writing – review and editing; Dexiang Zhu, Conceptualization, Investigation, Methodology, Writing – review and editing; Li Ren, Pengyang Zhou, Xuetong Yue, Investigation, Methodology; Ye Wei, Resources, Formal analysis, Writing – review and editing; Fuchu He, Conceptualization, Resources, Supervision, Methodology, Project administration; Jianmin Xu, Conceptualization, Resources, Funding

acquisition, Project administration, Writing – review and editing; Chen Ding, Conceptualization, Supervision, Funding acquisition, Investigation, Methodology, Writing – original draft, Project administration, Writing – review and editing

### Author ORCIDs
Aobo Zhuang  http://orcid.org/0000-0002-3958-7975
Chen Ding  http://orcid.org/0000-0001-8673-3464

### Ethics
The present study was carried out comply with the ethical standards of Helsinki Declaration II and approved by the Institution Review Board of Fudan University Zhongshan Hospital (B2019-166).

### Decision letter and Author response
Decision letter https://doi.org/10.7554/eLife.82959.sa1
Author response https://doi.org/10.7554/eLife.82959.sa2

---

## Additional files

### Supplementary files
• MDAR checklist

### Data availability
All data generated or analyzed during this study are included in the manuscript and supporting file. Source data files have been provided for all figures. The proteome raw data that support the findings of this study have been deposited to the ProteomeXchange Consortium (dataset identifier: PXD041476, https://proteomecentral.proteomexchange.org/cgi/GetDataset?ID=PXD041476) via the iProX partner repository (https://www.iprox.cn/) under Project ID IPX0003019000 at https://www.iprox.cn/page/project.html?id=IPX0003019000.

The following dataset was generated:

| Author(s) | Year | Dataset title | Dataset URL | Database and Identifier |
|---|---|---|---|---|
| Zhuang A, Zhuang A, Qin Z, Zhu D | 2023 | Proteomics Characteristics Reveal the Risk of T1 Colorectal Cancer Metastasis to Lymph Nodes | https://www.iprox.cn/page/project.html?id=IPX0003019000 | iprox, IPX0003019000 |

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
