## [Editor Report]

This paper seeks to answer an important clinical question by coming up with novel predictive biomarkers to predict high-risk T1 colorectal cancers that are at risk for nodal involvement with a machine-learning approach. The findings underscore that T1 CRC may have unique features and pathways that contributed to LNM.

---

## [Decision Letter]

**Decision letter after peer review:**

Thank you for submitting your article "Proteomics Characteristics Reveal the Risk of T1 Colorectal Cancer Metastasis to Lymph Nodes" for consideration by *eLife*. Your article has been reviewed by 3 peer reviewers, one of whom is a member of our Board of Reviewing Editors, and the evaluation has been overseen by Wafik El-Deiry as the Senior Editor. The reviewers have opted to remain anonymous.

Here are the overall comments and the essential revisions to consider. Please refer to Recommendations to Authors.

*Reviewer #1 (Recommendations for the authors):*

As a nonmass spec expert I can't comment on methods of development of the classifier but the clinical relevance and biomarker testing with training and validation subsets, one of which was prospective as well as the performance characteristics, makes the finding intriguing with clinical relevance with the ability to use IHC as well.

It was notable, that AUCs were compared using standard clinical features which demonstrated the classifier performed better.

This was a well-done paper asking clinical useful questions.

*Reviewer #2 (Recommendations for the authors):*

1. The authors are suggested to describe more on the relationship between proteomic characterizations of LNM-negative vs LNM-positive groups, mucinous colorectal adenocarcinoma with the biomarker exploration.

2. What is the rationale to define a training cohort and two different validation cohorts?

3. The authors are suggested to utilize additional CRC cohorts reported previously (Nature. 2014; 513:382; Cell. 2019; 177:1035; Cancer Cell. 2020; 38:734) for validating their 9 identified biomarkers.

4. The method described in this manuscript is confusing. For example, the proteins were extracted from tissues by Tris-lysis buffer, and the protein concentration was quantified by using Bradford assay. However, the authors further extracted the proteins from tissues by using an SDS-lysis buffer again.

5. The description of proteome data in this manuscript is confusing. An average of fewer than 6000 proteins were identified in each tissue sample, but totally more than 13000 proteins were identified from all samples. The total number is >2-fold higher than the average number of protein identification in each sample.

*Reviewer #3 (Recommendations for the authors):*

Overall the study shows a lot of potential – but many areas need improvement. It might be better published as a resource absent significant additional work.

In addition to standard genetics, correlation with a common oncogenic driver and tumor suppressor gene alterations would be helpful if available even for a subset of patients. The classifiers are also not tested on other, publicly available datasets, which might include genetics and later-stage patients, and patients with known outcomes to test whether these classifiers predict survival / DFS / PFS, etc.

The writing generally states that something meets a statistical threshold without stating the direction of change (for example, that there is a significant difference in the rate of metastasis between right and left colon would be better explained that right-sided CRCs metastasized less frequently than left-sided).

The section that generally describes the data is so general as to be unhelpful. For example in the section from ~317-346, no fold-changes are mentioned, so it is hard to know what something being upregulated or down-regulated means. The findings in coagulation and cytoskeleton are interesting in principle if more detail were present. The sentence "in conclusion, different differentiation statuses and histologic types in T1 CRC indicate specific pathways" is devoid of content.

The Discussion section is predominantly a summary of key results and would be better used contextualizing the work.

There is no meaningful difference between the two validation cohorts and they could be merged. The second validation cohort is described almost as if it were a clinical trial, but the data were not used in patient care. Figure 1 Figure Supplement 1 the legend is buried in the bottom corner and should be easier to find.

In figure 2C, the proteomaps are very pretty but devoid of information. These might be useful in contexts where there are large differences, but given the very small overall differences here, they are not helpful.

---

## [Author Response]

Reviewer #2 (Recommendations for the authors):1. The authors are suggested to describe more on the relationship between proteomic characterizations of LNM-negative vs LNM-positive groups, mucinous colorectal adenocarcinoma with the biomarker exploration.

We appreciate the reviewer’s comments. We agree with reviewer’s point that the relationship between proteomic characterizations of LNM-negative vs LNM-positive groups should be more described. Here's what we found in addition to our previous analysis (Figure 2 and Figure 2—figure supplement 1).

1) Comparison between our study and previous studies:

Firstly, we surveyed the published CRC LNM markers revealed in the literature. Among the 44 reported gene or RNA markers, only CTSD, GSTM3, and MX1 were differentially expressed between T1 CRC LNM-negative and LNM-positive patients in our proteomic data (Figure 2A). It indicates that the existing researches, including animal/cell models, gene/RNA related studies, etc., might not reflect the status of lymph node metastasis in T1 CRC. We also compared our data with those of the previous all stage CRC proteomic studies (Nature. 2014; 513:382 and Cancer Cell. 2020; 38:734) (PMID: 25043054, 32888432), 4,634 proteins were found in all three cohorts, whereas 2,577 proteins were detected specifically in our T1 CRC cohort (Figure 2—figure supplement 1A). These results suggest that compared with advanced CRC, T1 CRC may have its own unique protein patterns. These markers might improve the understanding of LNM in early-stage CRC.

2) The role of EMT for T1 CRC LNM:

To explore the biological processes associated with LNM in T1 CRC, we conducted gene set enrichment analysis (GSEA) to identify enriched pathways. The results revealed that only the epithelial-mesenchymal transition (EMT) pathway was significantly enriched (adjusted p-value < 0.05) in LNM-positive group, while no significantly enriched pathways in LNM-negative group (Figure 2C). We also examined the expression of EMT-related proteins and found that all of them were upregulated in the LNM-positive group compared to the LNM-negative group (log2FC >0; p<0.05, Wilcoxon rank-sum tests), with six proteins showing a significant upregulation (log2FC >1), including CAP2, PLOD1, SERPINE2, etc. (Figure 2E).

Next, we compared the differences in EMT marker expression between the LNM-positive and LNM-negative groups in all stages (T1-T4) CRC dataset. We found that the EMT markers that were significantly upregulated in the T1 CRC cohort showed no differences in either the CPTAC or mCRC cohorts (Figure 2F-H, Figure 2—figure supplement 1B,C). However, when we focused on the early-stage (T1/T2) CRC individuals from the mCRC cohort and compared the EMT marker expression between the LNM-positive and LNM-negative groups, the EMT markers like SERPINE2 and LRP1 were significantly upregulated in both our cohort and the early-stage CRC group from the mCRC cohort (log2FC >1 and p<0.05, Wilcoxon rank-sum test). Additionally, PLOD2 was also found to be up-regulated in LNM-positive group in T_1/2_ patients in mCRC cohort (Figure 2H). In summary, EMT might play an essential role during the process of LNM of tumor cells in early-CRC.

3) Differences in metabolic pathways between the LNM-positive and negative group:

We finally employed single sample Gene Set Enrichment Analysis (ssGSEA) in 166 significantly different expressed proteins (log2FC > 1 or <-1 and p<0.05, Wilcoxon rank-sum test) (Figure 2J). Amino acid metabolism pathways, such as "valine, leucine and isoleucine degradation" (hsa00280) and "glycine, serine and threonine metabolism" (hsa00260), were found to be enriched in the LNM-negative group (adjusted p ≤ 0.05, limma approach), and related proteins (TMLHE, SLC44A2, SHMT1 and EEF1E1) were up-regulated in the LNM-negative group (Log2FC <1, p < 0.05). (Figure 2K). On the other hand, the LNM-positive group mainly expressed lipid metabolism pathways, such as "arachidonic acid metabolism" (hsa00590) and "steroid biosynthesis" (hsa00100), which are known to promote cancer cell proliferation and migration and involved in the regulation of EMT. In addition, tumor metastasis-related signaling pathways, such as the MAPK and p53 pathways, as well as cellular process categories (NK cell mediated cytotoxicity and apoptosis), were enriched in LNM-positive patients. Meanwhile, the mTOR signaling pathway was enriched in the LNM-negative group. Overall, these findings suggest that different metabolic pathways and signaling pathways may contribute to LNM in T1 CRC.

In conclusion, unique protein markers specific to T1 CRC were found in this study, that may improve the understanding of LNM in early-stage CRC. EMT pathway was significantly enriched in the LNM-positive group of T_1/2_ CRC, and EMT-related proteins were found to be specifically up-regulated in the early-stage CRC LNM-positive group compared to the LNM-negative group. We also found differences in metabolic pathways and signaling pathways between the LNM-positive and negative groups, suggesting that different metabolic pathways and signaling pathways may contribute to LNM in T1 CRC.

We refined the analysis as shown in Figure 2 and Figure 2—figure supplement 1 in the revision, and we updated the relevant results and description in the “Proteomic Characteristics of the LNM-Negative and LNM-Positive Groups” part of Results.

2. What is the rationale to define a training cohort and two different validation cohorts?

We thank the reviewer for the critical comments. We sincerely apologized for the unclear description of our methodology. Our study aimed to address the clinical problem of whether additional surgical resection is needed after endoscopic resection and to accurately identify the rare but important LNM-positive patients in T1 CRC. To achieve this, we designed a rigorous study with multiple validation cohorts.

Training cohort:

The training cohort was a retrospective cohort consisting of 132 patients with LN dissection between June 2008 and October 2019. Due to the low probability of T1CRC LNM (3%-10%), we used propensity-score matching to reduce confounding biases.

In the training cohort, we established a proteomics-based lymph node metastasis prediction model for the first time, which can accurately predict T1 stage colorectal cancer lymph node metastasis, and is significantly better than the current guidelines.

Validation cohort 1:

The purpose of Validation Cohort 1 (VC1) was to validate the clinical applicability of our prediction model. We studied 42 consecutive samples during the same time period with train cohort (June 2008 to October 2019) in VC1. The AUC value of the model is 0.96 in VC1, which reflects the excellent universality of our model. It also verified the feasibility of applying our model to the prediction of lymph node metastasis in patients.

Validation cohort 2:

The validation cohort (VC2) is a prospective cohort, which is a unique aspect of our study design. After the model was developed in training cohort and validated in VC1 in the October 2019, we consecutively prospectively enrolled 47 samples over a six-month period (November 2019 to April 2020) in VC2. Importantly, VC2 was a single-blinded study, as we did not know the true pathological status before predicting it with our model.

To address the reviewer’ comments, we carefully, revised all the detailed descriptions in the “Materials and methods” section with the subtitle “Patient”.

3. The authors are suggested to utilize additional CRC cohorts reported previously (Nature. 2014; 513:382; Cell. 2019; 177:1035; Cancer Cell. 2020; 38:734) for validating their 9 identified biomarkers.

According to reviewer’s suggestion, we download the T_1/2_ CRC data from Bing Zhang’s (Cancer Cell. 2020, mCRC cohort) and Chen Li’s (Nature. 2014, CPTAC cohort) studies, and validated our 9-protein simplified classifier in these two data sets, respectively.

Firstly, we screened the data according to our inclusion criteria. Since the two external validation datasets focused on the all-stage CRC, the number of patients in the T1 stage was small, 6 patients for mCRC cohort and 3 for CPTAC cohort, we added T2NxM0 patients to the model validation.

We employed ComBat, an Empirical Bayes method, to reduce the batch effects between our data set and the two data sets mCRC cohort and CPTAC cohort (Figure 3—source data 4). After batch correction, nine proteins from our simplified classifier were selected, the prediction score was calculated for each sample from both mCRC cohort and CPTAC cohort, respectively (Figure3-source data 4). Consistent with our previous analyses, 0.420 was regarded as a cutoff value, when stratifying the patients into “high-risk” and “low-risk” groups by predicting risk score for LNM range from 0 to 1, and at this threshold, all patients in the low-risk group were LNM-negative.

In mCRC cohort (T_1/2_ CRC patients, N=16; LNM-, N=13; LNM+, N=3), 9 LNM-negative patients were classified into low-risk group and all the patient stratified into the low-risk group are LNM-negative, corresponding to a sensitivity of 100% and a specificity of 70% (Figure 3I).

In CPTAC cohort (T_1/2_ CRC patients, N=16; LNM-, N=15; LNM+, N=1), 11 out of 15 LNM-negative patients were correctly identified, corresponding a specificity of 73%. Although, the results were limited by the number of LNM-positive patients, LNM-positive patients were successfully assigned to the high-risk group with a sensitivity of 100% (Figure 3J).

These results indicated that our model was able to effectively identify patients with LNM during external validation, both in mCRC cohort and CPTAC cohort, ensuring that no patients with metastasis were missed. At the same time, compared with the NCCN guidelines, our model classifies more patients without metastasis into the low-risk group, reducing the incidence of overtreatment, and would provide a valuable insight for clinical decisions to T1 CRC patients treatment.

In the revision, we have updated Figure 3 and Figure 3—figure supplement 1, and the “External validation of the Simplified classifier” section of the “Result”.

4. The method described in this manuscript is confusing. For example, the proteins were extracted from tissues by Tris-lysis buffer, and the protein concentration was quantified by using Bradford assay. However, the authors further extracted the proteins from tissues by using an SDS-lysis buffer again.

Thank the reviewer for the comments. We apologize for the unclear description of proteins extraction process. We have carefully checked and revised the “Materials and methods” section.

The formulation of the SDS lysis buffer was 4% SDS, 0.1 M Tris-HCl (pH 8.0), 0.1 M DTT and 1 mM PMSF. We first added the buffer to the extracted tissues before adding SDS to the solution, for the sample needs to be sonicated and the high concentration of SDS will create a lot of foam. After the sample was sonicated, the supernatants were collected, and the protein concentration was determined using the Bradford assay. Then, due to the sample would later be tested by MS, SDS was added and kept for 2.5 h at 99 °C with shaking at 1800 rpm to get rid of impurities such as protein-binding-DNA et al.

In the revision, we have revised this section in the “Materials and methods”. The details were shown as follows.

Original version: “Lysis buffer [0.1 M Tris-HCl (pH 8.0), 0.1 M DTT (Σ, 43815), 1 mM PMSF (Amresco, M145)] was added to the extracted tissues, and subsequently sonicated for 1 min (3 s on and 3 s off, amplitude 25%) on ice. The supernatants were collected, and the protein concentration was determined using the Bradford assay. The extracted tissues were then lysed with 4% sodium dodecyl sulfate (SDS) and kept for 2.5 h at 99 °C with shaking at 1800 rpm.”

Revised version: “Lysis buffer [0.1 M Tris-HCl (pH 8.0), 0.1 M DTT (Σ, 43815), 1 mM PMSF (Amresco, M145)] was added to the extracted tissues before adding SDS to the solution. The solution with the samples were sonicated for 1 min (3 s on and 3 s off, amplitude 25%) on ice. The supernatants were collected, and the protein concentration was determined using the Bradford assay. 4% sodium dodecyl sulfate (SDS) was added and kept for 2.5 h at 99 °C with shaking at 1800 rpm.”

5. The description of proteome data in this manuscript is confusing. An average of fewer than 6000 proteins were identified in each tissue sample, but totally more than 13000 proteins were identified from all samples. The total number is >2-fold higher than the average number of protein identification in each sample.

Thanks for the comments. We apologize for the unclear description of proteins identification process.

In our study, firstly, peptide identification stringency was set at a maximum 1% FDR at peptide level, and then proteins with 1% FDR were select. As a result, we identified 13,091 proteins with an average more than 5,000 proteins in the tumor samples. The same cutoff strategies of FDR at protein/peptide level have been widely used in recently published researches (PMID: 34534465, PMID: 33577785). Moreover, to increase reliability, not all identified proteins were used for downstream data analysis. Proteins for further analysis were required to have at least 1 unique peptide and more than 10^−5^ in FOT, we also excluded keratins to ensure the credibility of the data (PMID: 29520031). Applying this cut-off value, we used the remaining 9,402 proteins for following analysis. Furthermore, during the subsequent data analysis, the proteins of each group should first meet the criteria with expression frequency of more than 30% patients in the group as shown in Figure 1—figure supplement 2H .

The number, frequency and coverage depth of proteins identified in our study are consistent with those reported in previous published studies. For instance, the colon and rectal cancer study from CPTAC published in Nature (PMID: 25043054), identified 7,526 proteins from 95 tumor samples with average of 4,845 proteins in each sample. The metastatic colorectal cancer study published in Cancer Cell (PMID: 32888432), identified 11,510 proteins from 145 tissue samples with average of 5,715 proteins. The early gastric cancer cohort (PMID: 34818622) identified 15,158 proteins with average of 5,113 from 324 samples.

In the revision, we have updated Figure 1—figure supplement 2 and the “Peptide and protein identification” section of the “Materials and methods”.

Reviewer #3 (Recommendations for the authors):Overall the study shows a lot of potential – but many areas need improvement. It might be better published as a resource absent significant additional work.In addition to standard genetics, correlation with a common oncogenic driver and tumor suppressor gene alterations would be helpful if available even for a subset of patients.

Thanks for reviewer’s suggestion, we appreciate it very much. We agree that genetic testing plays an important role in the diagnosis and treatment of colorectal cancer. However, its significance in the lymph node metastasis of early-stage colon cancer is still unclear. In our previous version, we did not perform an examination of genomics due to sample number and size limitations. According to reviewer’s advices, we have collaborated with our pathology colleagues to perform additional genetic testing. The testing mainly included microsatellite status and *RAS*, *RAF* and *PIK3CA* gene status, and improved our study in two parts.

MSI status

To test MSI status, we performed IHC to evaluate the expression of 4 mismatch repair (MMR) proteins (MLH1, MSH2, MSH6 and PMS2) in 171 of 221 (77.4%) patients in our cohort (Author response image 1, Figure1B and Figure1-source data1). We tested the MSI status by using IHC staining, a commonly used method in CRC MSI status clinical test (PMID: 21497289). The patients showed positive staining of the nuclei of all four MMR proteins were considered as proficient MMR (pMMR)/ microsatellite stable (MSS), and the mismatch repair-deficient (dMMR) /microsatellite instability-high (MSI-H) cases show the loss of one of the two MLH1/PMS2 or MSH2/MSH6 heterodimers (Materials and methods in the revision) (PMID: 25560596). There were 12 (7%) MSI-H patients in our study, this proportion is consistent with previous reported on metastatic CRC by Federico et al. (6.6%) (PMID: 30865548). For MSI-H patients, there were 16.7% (2 of 12) had LNM, and 39.6% (63 of 159) for MSS patients, the MSS group showed a higher tendency of LNM (Figure 1—figure supplement 1K). Our results are in agreement with the previous studies (PMID: 25510762). Further analysis of our T1 CRC proteomic data found that the MLH1 was significantly up-regulated in LNM-positive group compared with LNM-negative group (Log2FC = 1.28; p = 0.01, Wilcoxon rank-sum test) (Author response image 1).

Gene mutations

We detected 3 common gene mutations (*RAS, BRAF* and *PIK3CA*) associated with prognosis in CRC in 132 patients of our studies using PCR (Materials and methods in the revision). The results showed that 70 of 132 (53%) individuals had no mutations in all 3 genes (Figure1B and Figure1-source data1). There were 48 (36.4%) individuals had mutations in the *KRAS*-exon2; 4 (3%) had *KRAS*- exon3 mutations; 2 (1.5%) had *KRAS*-exon4 mutations; 2 (1.5%) had *NRAS* mutations; 4 (3%) had mutations in *BRAF* V600E; 3 (2.3%) had *PIK3CA* mutations. There was one patient has mutations in both *NRAS, PIK3CA* and *KRAS*-exon2. The mutation ratio was similar to previous studies (PMID: 25911860). Nineteen of 62 (30.6%) patients with gene mutations and 26 of 70 (37.1%) patients without mutations had LNM, indicate there was no statistical difference between the two groups (p=0.432) (Author response image 1).

In the revision, we have updated Figure 1 and Figure 1—figure supplement 1 and the “Materials and methods” and “Result” section.

**Author response image 1. sa2fig1:** (A) Heatmap of Clinical parameters, MSI and gene mutations (*RAS*, *BRAF* and *PIK3CA*). (B) Comparison of MLH1 expression between LNM-negative and positive groups. The p value was calculated by Wilcoxon rank-sum test.

The classifiers are also not tested on other, publicly available datasets, which might include genetics and later-stage patients, and patients with known outcomes to test whether these classifiers predict survival / DFS / PFS, etc.

Thank the reviewer for the comments. According to reviewer’s suggestion, we download the T_1/2_ CRC data from Bing Zhang’s (Cancer Cell. 2020, mCRC cohort) and Chen Li’s (Nature. 2014, CPTAC cohort) studies, and validated our 9-protein simplified classifier in these two data sets, respectively.

Firstly, we screened the data according to our inclusion criteria. Since the two external validation datasets focused on the all-stage CRC, the number of patients in the T1 stage was small, 6 patients for mCRC cohort and 3 for CPTAC cohort, we added T2NxM0 patients to the model validation.

We employed ComBat, an Empirical Bayes method, to reduce the batch effects between our data set and the two data sets mCRC cohort and CPTAC cohort (Figure 3—source data 4 ). After batch correction, nine proteins from our simplified classifier were selected, the prediction score was calculated for each sample from both mCRC cohort and CPTAC cohort, respectively (Figure3-source data 4 in the revision). In this study, 0.420 was regarded as a cutoff value, when stratifying the patients into “high-risk” and “low-risk” groups by predicting risk score for LNM range from 0 to 1, and at this threshold, all patients in the low-risk group were LNM-negative.

In mCRC cohort (T_1/2_ CRC patients, N=16; LNM-, N=13; LNM+, N=3), 9 LNM-negative patients were classified into low-risk group and all the patient stratified into the low-risk group are LNM-negative, corresponding to a sensitivity of 100% and a specificity of 70% (Figure 3I).

In CPTAC cohort (T_1/2_ CRC patients, N=16; LNM-, N=15; LNM+, N=1), 11 out of 15 LNM-negative patients were correctly identified, corresponding a specificity of 73%. However, the results were limited by the number of LNM-positive patients, and LNM-positive patients were successfully assigned to the high-risk group with a sensitivity of 100% (Figure 3J).

These results indicated that our model was able to effectively identify patients with LNM during external validation, both in mCRC cohort and CPTAC cohort, ensuring that no patients with metastasis were missed. At the same time, compared with the NCCN guidelines, our model classifies more patients without metastasis into the low-risk group, reducing the incidence of overtreatment, and would provide a valuable insight for clinical decisions to T1 CRC patients treatment.

In the revision, we have updated Figure 3 and Figure 3—figure supplement 1, and the “External validation of the Simplified classifier” section of the “Results” in the revised manuscript.

The writing generally states that something meets a statistical threshold without stating the direction of change (for example, that there is a significant difference in the rate of metastasis between right and left colon would be better explained that right-sided CRCs metastasized less frequently than left-sided).

We thank the reviewer for the critical comments. We sincerely apologized for the unclear presentation of direction of changes. The chi-square test is used in our study to compare difference between two or more categorical variables, and p<0.05 considered is to indicate a significant difference, the group with a higher percentage is more likely to have a higher event occurrence rate (LNM in our study). To address the reviewer’ comments, we carefully, revised all the detailed descriptions in the manuscript. For example:

“In our cohort, the rate of LNM was related to tumor location, and the left-sided tumors (LNM rate: 41.1%) showed a higher metastatic tendency than right-sided (LNM rate: 22.0%) (p=0.036, chi-square test).”

“In agreement with previous reports [5], in our T1 CRC cohort the ratio of LNM was significantly higher in patients with poorly differentiated patients (48.6%) compared with well-moderately differentiated patients, in T1 CRC (28.8%) (p=0.004, chi-square test), and mucinous adenocarcinoma patients (55%) also shown a greater tendency to LNM than adenocarcinoma (30.1%) (p=0.004, chi-square test)”

The section that generally describes the data is so general as to be unhelpful. For example in the section from ~317-346, no fold-changes are mentioned, so it is hard to know what something being upregulated or down-regulated means. The findings in coagulation and cytoskeleton are interesting in principle if more detail were present. The sentence "in conclusion, different differentiation statuses and histologic types in T1 CRC indicate specific pathways" is devoid of content.

Thank the reviewer for the comments. Here, we divided the response into three parts to answer: (1) Describe the data in detail; (2) details of findings in coagulation and cytoskeleton; (3) the sentence "in conclusion……” is devoid of content.

1) Describe the data in detail:

The proteome analysis results were uniformly defined as 'up-regulated' or 'down-regulated' based on the measurements from a statistical test (two-sided Student’s t-test/Wilcoxon rank-signed test/ Kruskal-Walli’s test) and the log2 transformed fold change (≥ 1 for upregulation and ≤ -1 for downregulation). According to the reviewer’s advice, we have revised the detailed descriptions of the fold-changes of proteins in the “Result” section. For example:

“We found that the EMT markers that were significantly up-regulated (log2FC >1) in the T1 CRC cohort showed no differences in either the CPTAC or mCRC cohorts.”

“We also examined the expression of EMT-related proteins and found that all of them were up-regulated in the LNM-positive group compared to the LNM-negative group (log2FC >0; p<0.05, Wilcoxon rank-sum tests).”

“Cytoskeletal remodeling proteins, including ABI1, SPTA1, SPTB, ANK1, MRPL46 and ITGA2B, were down-regulated in patients with LNM compared with those without LNM (log2FC <-1 and p<0.05, Wilcoxon rank-sum test).”

“RHOT2 was significant down-regulated (Log2FC=-1.35; p=0.003, Wilcoxon rank-sum test) in LNM-positive patients compared with LNM-negative patients in our T1 CRC cohort.”

2) Details of findings in coagulation and cytoskeleton:

As shown in Figure 2, in our previous analysis, we found the proteins overexpressed by LNM-negative patients were involved in cytoskeletal remodeling (log2FC <-1 and p<0.05, Wilcoxon rank-sum test), including SPTA1, SPTB, ANK1, MRPL46 and ITGA2B. The cytoskeletal remodeling related pathway (GO:0003774) was found to be up-regulated in LNM-negative patients of mCRC cohort (Figure 2—figure supplement 1D). Meanwhile, twelve of the 84 proteins that were elevated in LNM-positive patients (log2FC > 1 and p<0.05, Wilcoxon rank-sum test) were related to coagulation cascades, including CAP2, CEACAM6, F13A1, GNG2, GPC1, GUCY1B3, ISLR, MAGED2, SERPINE2, SERPINB8, SHC1 and VWF.

As shown in Figure 2—figure supplement 1H, further analysis reveals the relationship between cytoskeletal remodeling and coagulation cascades. As expected, the expression of cytoskeletal remodeling related proteins including SPTA1, SPTB, ANK1 and ITGA2B showed positive correlation in our study (P < 0.05, coefficient >0, Spearman), as did coagulation cascades related proteins. Meanwhile, we found the negative correlations between the cytoskeletal remodeling related proteins and coagulation cascades, for example, the expression of SPTA1 was negative correlated with 7 coagulation cascades related proteins including CEACAM6, MAGED2, ISLR, GPC1, SHC1, SERPINB8 and GNG2, MRPL46 showed a negative correlation with 5 coagulation cascades related proteins. ANK1 and SPTB were also found to be negatively correlated with coagulation cascades. Recent studies have shown that cytoskeletal remodeling can affect complement and coagulation pathway activation. SPTA1 and SPTB are components of the erythrocyte cytoskeleton, and defects in these proteins can lead to hemolytic anemia, which can activate the coagulation cascade. MRPL46 is a mitochondrial ribosomal protein that may play a role in mitochondrial function, which can affect coagulation and inflammation.

In conclusion, the alteration of cytoskeletal remodeling and coagulation cascades pathways in T1 CRC and their interaction may affect LNM.

3) The sentence "in conclusion……” is devoid of content:

Thanks for the comment. We apologize for not summarizing the results effectively. The sentence was revised as follow:

“In conclusion, oxidative phosphorylation and TCA cycle pathways were enriched in the well to moderately differentiated adenocarcinoma subgroup, while GTPase activity and Wnt signaling pathways were enriched in poorly differentiated adenocarcinoma. The mucinous adenocarcinoma subgroup was characterized by aggressive pathways such as ECM organization, cell migration, and vesicle-mediated transport.”

The Discussion section is predominantly a summary of key results and would be better used contextualizing the work.

Thank the reviewer for the valuable comments. We apologize for not appropriately contextualizing the “Discussion” section. In the revised manuscript, we rewrite the discussion part as follow:

“Here, we present a comprehensive proteomic study to focus on LNM in patients with submucosal T1 CRCs. Using a mass spectrometry-based proteomic technique, we analyzed 221 T1 CRC patients and identified different molecular characteristics between patients with and without LNM. We also uncovered protein characteristics of mucinous colorectal adenocarcinoma.

Based on the T1 CRC proteomics dataset, we developed a high-performing protein signature-based model for predicting LNM, which can aid in making informed decisions regarding additional surgical resection following endoscopic resection of T1 CRCs. The protein signatures successfully stratified patients according to their risk of LNM. In the training cohort, 105 candidate protein markers were reduced to 55 potential predictors by examining the predictor-outcome association with the LASSO method. Based on these 55 biomarkers, 10-fold cross-validation was applied to the training cohort and generated an ROC curve. The classifier achieved an AUC of 1.00 in the training cohort, which was confirmed in two independent validation cohorts. The 55 protein markers achieved an AUC of 0.96 (95% CI, 0.917 to 1.000) in the ESD validation cohort and 0.933 (95% CI, 0.858 to 1.000) in the prospectively collected validation cohort. Besides, a simplified model consisting of only 9 proteins, which could effectively identify patients with LNM in T_1/2_ CRC, was developed and validated on external data.

In recent years, there have been many studies on LNM in T1 CRC, among which the following studies are representative: in 2021, Shin-Ei Kudo and colleagues developed a machine-learning artificial neural network using data on patients' age and sex, tumor characteristics, and histologic grade to identify patients with T1 CRCs who had LNM, the ANN model outperformed guidelines in identifying high-risk patients who require radical surgery. In 2022, Yinghui Zhao and colleagues conducted genome-wide methylation profiling of T1 CRC specimens and developed a nine-CpG signature that can distinguish LNM-positive versus LNM-negative specimens and pre-treatment biopsies. The signature has potential to improve selection of high-risk patients. Also in 2022, Yuma Wada and colleagues developed a blood-based liquid biopsy assay to detect LNM in high-risk submucosal T1 CRCs. Their transcriptomic panel of 4 miRNAs and 5 mRNAs showed robust identification of patients with LNM. Compared to the above-mentioned studies, in addition to higher predictive accuracy, this study has the following advantages: since the ultimate goal of the project was to precisely identify the LNM in T1 CRC, and to reduce additional surgery after endoscopic resection for LNM-negative patients, we set up a cohort only containing ESD samples to validate our predictive model, and a single-blinded prospective cohort was also used to validate the accuracy of the model. Furthermore, we demonstrated the potential for clinical translation of our study. Base on proteomic result, we further constructed an effective classifier using five proteins based on their IHC score to optimize our model for clinical use.

In this study, to ensure the accuracy of LN status of the enrolled patients, the dissected number of LN in all patients including both surgical resection and ESD was more than 12. However, the longer-term follow-up data, including DFS, PFS, etc., are not available due to limitations in sample collection time and the prognosis of such patients needs to be tracked over long periods of time, may impact the strength of our conclusions. To address this limitation, we used propensity-score matching to reduce confounding biases in our training cohort. Patients were prospectively enrolled in our validation cohort, which was designed as a single-blinded prospective study to enhance the rigor and reliability of our findings. Furthermore, the presence of isolated tumor cells (ITCs) or micrometastases (MMs) within regional LN are not considered, due to conventional histopathologic examination cannot detected them. According to previous studies, there were about 5% pT1N0 gastric cancer patients have ITCs in LN, and 10% in pT1Nx CRC. The effect of MMs on prognosis in pT1N0 CRC is still unclear. The present of ITCs/MMs in LN may explain why there are nearly 13% (29 of 221) LNM-negative patients were classified into high-risk group by the prediction model in our study. We will continuously follow the prognosis of the patients, and the ITCs/MMs in LN also need to be further validated in the future studies.”

Besides, we found dysregulation of RHOT2 expression can impact LNM in T1 CRC. Although the function of RHOT2 in cancer is still unclear, the expression of its paralog RHOT1 affects metastasis in a variety of tumors, including pancreatic cancer, gastric cancer, small cell lung cancer, etc. In addition, previous studies have found that Myc regulation of mitochondrial trafficking through RHOT1 and RHOT2 enables tumor cell motility and metastasis. In our research, we demonstrated that low expression of RHOT2 was associated with LNM and poor prognosis, and knocking down RHOT1 can significantly increase migration of CRC cells. The mechanism by which RHOT2 affects CRC LNM is not fully understood, but may involve several pathways. One potential pathway involves the regulation of EMT, a process by which cancer cells acquire invasive properties and become more motile. More analytical studies and experiments are needed in our future researches to understand the specific role and mechanism of RHOT2 in the process of tumor metastasis.

In conclusion, we used proteomic analysis to identify molecular characteristics and develop the high accuracy protein signature-based models for predicting LNM in patients with T1 CRC.

There is no meaningful difference between the two validation cohorts and they could be merged. The second validation cohort is described almost as if it were a clinical trial, but the data were not used in patient care.

We thank the reviewer for the critical comments. We sincerely apologized for the unclear description of our methodology. Our study aimed to address the clinical problem of whether additional surgical resection is needed after endoscopic resection and to accurately identify the rare but important LNM-positive patients in T1 CRC. To achieve this, we designed a rigorous study with multiple validation cohorts.

Training cohort:

The training cohort was a retrospective cohort consisting of 132 patients with LN dissection between June 2008 and October 2019. Due to the low probability of T1CRC LNM (3%-10%), we used propensity-score matching to reduce confounding biases.

In the training cohort, we established a proteomics-based lymph node metastasis prediction model for the first time, which can accurately predict T1 stage colorectal cancer lymph node metastasis, and is significantly better than the current guidelines.

Validation cohort 1:

The purpose of Validation Cohort 1 (VC1) was to validate the clinical applicability of our prediction model. We studied 42 consecutive samples during the same time period with train cohort June 2008 to October 2019 in VC1. The AUC value of the model is 0.96 in VC1, which reflects the excellent universality of our model. It also verified the feasibility of applying our model to the prediction of lymph node metastasis in patients.

Validation cohort 2:

The validation cohort (VC2) is a prospective cohort, which is a unique aspect of our study design. After the model was developed in training cohort and validated in VC1 in the October 2019, we consecutively prospectively enrolled 47 samples over a six-month period (November 2019 to April 2020) in VC2. Importantly, VC2 was a single-blinded study, as we did not know the true pathological status before predicting it with our model.

To address the reviewer’ comments, we carefully, revised all the detailed descriptions in the “Materials and methods” section with the subtitle “Patient”.

Figure 1 Figure Supplement 1 the legend is buried in the bottom corner and should be easier to find.

Thank the reviewer for the comments. We have made the figure legend more visible and easier to find in Figure 1-figure supplement 1,

In figure 2C, the proteomaps are very pretty but devoid of information. These might be useful in contexts where there are large differences, but given the very small overall differences here, they are not helpful.

We appreciate the reviewer’s comments. We agree with reviewer’s point that the relationship between proteomic characterizations of LNM-negative vs LNM-positive groups should be more described. Here's what we found in addition to our previous analysis (Figure 2).

1) Comparison between our study and previous studies:

Firstly, we surveyed the published CRC LNM markers revealed in the literature. Among the 44 reported gene or RNA markers, only CTSD, GSTM3, and MX1 were differentially expressed between T1 CRC LNM-negative and LNM-positive patients in our proteomic data (Figure 2A). It indicates that the existing researches, including animal/cell models, gene/RNA related studies, etc., might not reflect the status of lymph node metastasis in T1 CRC. We also compared our data with those of the previous all stage CRC proteomic studies (Nature. 2014; 513:382 and Cancer Cell. 2020; 38:734) (PMID: 25043054, 32888432), 4,634 proteins were found in all three cohorts, whereas 2,577 proteins were detected specifically in our T1 CRC cohort (Figure 2B). These results suggest that compared with advanced CRC, T1 CRC may have its own unique protein patterns. These markers might improve the understanding of LNM in early-stage CRC.

2) The role of EMT for T1 CRC LNM:

To explore the biological processes associated with LNM in T1 CRC, we conducted gene set enrichment analysis (GSEA) to identify enriched pathways. The results revealed that only the epithelial-mesenchymal transition (EMT) pathway was significantly enriched (adjusted p-value < 0.05) in LNM-positive group, while no significantly enriched pathways in LNM-negative group (Figure 2C). We also examined the expression of EMT-related proteins and found that all of them were upregulated in the LNM-positive group compared to the LNM-negative group (log2FC >0; p<0.05, Wilcoxon rank-sum tests), with six proteins showing a significant upregulation (log2FC >1), including CAP2, PLOD1, SERPINE2, etc. (Figure 2D).

Next, we compared the differences in EMT marker expression between the LNM-positive and LNM-negative groups in all stages (T1-T4) CRC dataset. We found that the EMT markers that were significantly upregulated in the T1 CRC cohort showed no differences in either the CPTAC or mCRC cohorts (Figure 2E-I). However, when we focused on the early-stage (T1/T2) CRC individuals from the mCRC cohort and compared the EMT marker expression between the LNM-positive and LNM-negative groups, we found two markers like SERPINE2 and LRP1, that were significantly upregulated in both our cohort and the early-stage CRC group from the mCRC cohort (log2FC >1 and p<0.05, Wilcoxon rank-sum test). Additionally, PLOD2 was also found to

be up-regulated in LNM-positive group in T_1/2_ patients of mCRC cohort (Figure 2G). In summary, EMT might play an essential role during the process of LNM of tumor cells in early-CRC.

3) Differences in metabolic and other pathways between the LNM-positive and negative group:

We finally employed single sample Gene Set Enrichment Analysis (ssGSEA) in 166 significantly different expressed proteins (log2FC > 1 or <-1 and p<0.05, Wilcoxon rank-sum test) (Figure 2J). Amino acid metabolism pathways, such as "valine, leucine and isoleucine degradation" (hsa00280) and "glycine, serine and threonine metabolism" (hsa00260), were found to be enriched in the LNM-negative group (adjusted p ≤ 0.05, limma approach), and related proteins (TMLHE, SLC44A2, SHMT1 and EEF1E1) were up-regulated in the LNM-negative group (Log2FC <1, p < 0.05). (Figure 2K). On the other hand, the LNM-positive group mainly expressed lipid metabolism pathways, such as "arachidonic acid metabolism" (hsa00590) and "steroid biosynthesis" (hsa00100), which are known to promote cancer cell proliferation and migration and involved in the regulation of EMT. In addition, tumor metastasis-related signaling pathways, such as the MAPK and p53 pathways, as well as cellular process categories (NK cell mediated cytotoxicity and apoptosis), were enriched in LNM-positive patients. Meanwhile, the mTOR signaling pathway was enriched in the LNM-negative group. Overall, these findings suggest that different metabolic pathways and signaling pathways may contribute to LNM in T1 CRC.

In conclusion, unique protein markers specific to T1 CRC were found in this study, that may improve the understanding of LNM in early-stage CRC. EMT pathway was significantly enriched in the LNM-positive group of T_1/2_ CRC, and EMT-related proteins were found to be specifically up-regulated in the early-stage CRC LNM-positive group compared to the LNM-negative group. We also found differences in metabolic pathways and signaling pathways between the LNM-positive and negative groups, suggesting that different metabolic pathways and signaling pathways may contribute to LNM in T1 CRC.

We refined the analysis as shown in Figure 2 and Figure 2—figure supplement 1 in the revision, and we updated the relevant results and description in the “Proteomic Characteristics of the LNM-Negative and LNM-Positive Groups